

# Snowpack modelling in the Pyrenees driven by kilometric resolution meteorological forecasts

Louis Quéno[1], Vincent Vionnet[1], Ingrid Dombrowski-Etchevers[2], Matthieu Lafaysse[1], Marie Dumont[1], and Fatima Karbou[1]

[1]Météo-France/CNRS, CNRM - GAME UMR3589, CEN, St. Martin d'Hères, France
[2]Météo-France/CNRS, CNRM - GAME UMR3589, Toulouse, France

*Correspondence to:* L. Quéno (louis.queno@meteo.fr)

**Abstract.** Distributed snowpack simulations in the French and Spanish Pyrenees were carried out using the detailed snowpack model Crocus driven by the Numerical Weather Prediction system AROME at 2.5 km grid spacing, during four consecutive winters, from 2010 to 2014. The aim of this study was to assess the benefits of kilometric resolution atmospheric forcing to a snowpack model

for describing the spatial variability of seasonal snow cover within a mountain range. The evaluation was performed through comparisons to ground-based measurements of snow depth, snow water equivalent and precipitation, to satellite snow cover images and to snowpack simulations driven by the analysis system SAFRAN. Snow depths simulated by AROME–Crocus exhibit an overall positive bias, particularly marked on the first summits near the Atlantic Ocean. The simulation of

mesoscale orographic effects by AROME allows to capture a realistic regional snowpack variability, unlike SAFRAN–Crocus. The categorical study of daily snow depth variations gives a differentiated perspective of accumulation and ablation processes. Both models underestimate strong snow accumulations and strong snow depth decreases, which is mainly due to the non-simulated wind-induced erosion, the underestimation of strong melting and an insufficient settling after snowfalls. The prob-

lematic assimilation of precipitation gauge measurements is also emphasized, which raises the issue of a dedicated analysis to complement the benefits of AROME kilometric resolution and dynamical behaviour in mountainous terrain.

## 1 Introduction

A major challenge of seasonal snow cover studies in mountainous terrain is to take into account the

high spatial variability of the snowpack, since it affects a large number of phenomena in mountains. It is particularly necessary for avalanche hazard forecasting or mountain hydrology. Snow cover heterogeneous distribution is indeed the main factor controlling the runoff during the melting season (Anderton et al., 2002), as well as an essential factor of avalanche formation (Schweizer et al., 2003). Seasonal snow heterogeneity also has a wide range of effects on the different plant species of the





alpine tundra (Jonas et al., 2008b), and crucially constrains animal life at high altitude (Jonas et al., 2008a).

The spatial variability of the snowpack is observed at different scales and is mainly caused by the spatial variability of atmospheric conditions, on the same range of scales. The regional climate determines the main synoptic weather patterns that contribute to building up the snow cover. Within

a mountain range, the snowpack spatial variability is caused at a given elevation by enhanced or reduced local exposure to the synoptic flows bringing snowfall. Additionally, the atmospheric conditions at the surface vary following local topography, e.g. elevation influence on temperature, precipitation phase and radiations, and slope and aspect influences on incoming solar radiation. At a smaller scale (less than 100m), processes like wind-induced erosion (Pomeroy and Gray, 1995) or

preferential deposition of snowfall on the leeward slopes (Lehning et al., 2008), play a decisive role on snow distribution (e.g. Mott et al., 2010).

The description of the snowpack variability through snowpack modelling is thus highly dependent on the spatial resolution of the atmospheric forcing used. This variability is currently represented by classes of elevation, slope and aspect at a scale of about 1000 km$^2$ for operational avalanche haz-

ard forecasting in French mountainous areas. The detailed snowpack model SURFEX/ISBA/Crocus (Vionnet et al., 2012), mentioned as Crocus hereafter, is used within the SAFRAN–SURFEX/ISBA/ Crocus–MEPRA model chain (Durand et al., 1999; Lafaysse et al., 2013). The meteorological analysis and forecasting system SAFRAN (Système d'Analyse Fournissant des Renseignements Atmosphériques à la Neige; Analysis System Providing Atmospheric Information to Snow; Durand et al.,

1993) provides relevant meteorological parameters affecting snowpack evolution, with a dependence on the elevation within mountain ranges, so called "massifs", assumed to be meteorologically homogeneous. SAFRAN was also used in many other applications such as the study of snow cover climatology in the French Alps from 1958 to 2005 (Durand et al., 2009a, b).

The atmospheric forcing of snowpack models for distributed simulations (i.e. on a regular grid) has

been recently the focus of many studies, building on the development of NWP (Numerical Weather Prediction) models of increasing resolution. Bellaire et al. (2011, 2013) performed snowpack simulations in Canada with the detailed snow cover model SNOWPACK (Bartelt and Lehning, 2002), driven by the 15 km resolution regional NWP model GEM15 (Mailhot et al., 2006), in the aim of avalanche hazard forecasting. They highlighted the potential of distributed snow cover simulations

driven by NWP systems, as a useful complement of snow cover observations in regions where such observations are lacking. For snowpack simulations in mountainous terrain, kilometric atmospheric information allows to capture an important part of intra-massif snowpack variability. Such simulations were performed by Bellaire et al. (2014) in New-Zealand for avalanche hazard forecasting, driving SNOWPACK by the NWP model ARPS (Advanced Regional Prediction System, Xue et al.,

2000) at 3 km and 1 km horizontal resolution. This study shows better results in terms of snowfall for the highest resolution forcing over a 10 days snowy period. Horton et al. (2015) demonstrated



the benefits of forcing SNOWPACK with the 2.5 km resolution NWP model GEM-LAM (Erfani et al., 2005) for specific studies of snowpack stability (surface hoar layers formation). Schirmer and Jamieson (2015) applied the same chain of models GEM-LAM/SNOWPACK in the mountains of
western Canada and north-western US, with a focus on winter precipitation, and showed the better performance of the kilometric resolution NWP system with respect to GEM15 (15 km) and a precipitation analysis system, particularly in terms of snowfall quantitative distribution. Snowpack variability can also be simulated at scales of tens of meters, using adequate snowpack-atmosphere coupled models. Vionnet et al. (2014) used the coupled system Meso-NH/Crocus for studying wind-
induced erosion of the snowpack, at a 50 m horizontal resolution; and Mott et al. (2014) used the atmospheric model ARPS at a 75 m horizontal resolution for studying the orographic effects on snow deposition patterns. Such simulations can only be done on very limited areas, due to obvious computing limitations, and cannot currently be applied for operational issues like avalanche hazard forecasting or mountain hydrology.

The aim of the present study is to simulate snowpack variability within a whole mountainous chain. Consequently, kilometric snowpack simulations offer a promising compromise between spatial resolution and computational time. AROME (Application of Research to Operations at MEsoscale, Seity et al., 2011) is a NWP system, providing operational short-range numerical weather forecasts over France at 2.5 km grid spacing since December 2008. Its kilometric resolution over the French
mountains offers an alternative to the forcing of Crocus by SAFRAN, at higher resolution, but without a dedicated analysis system. AROME has been preliminarily evaluated in mountainous terrain by Dombrowski-Etchevers et al. (2013) and Vionnet et al. (2015b), who showed its skill for mountain meteorological forecast in the French Alps. Vionnet et al. (2015b) discussed the potential of AROME–Crocus for snowpack modelling in the French Alps. They illustrated the realistic represen-
tation of the intra-massif spatial variability of the snowpack for this region, although the improved resolution does not compensate the lack of a dedicated analysis system. Subsequently, this paper proposes to extend the study over the French and Spanish Pyrenees, where the behaviour might be different because of the particular climate of the Pyrenees influenced by the vicinity of the Atlantic Ocean and the Mediterranean Sea. We also present a refinement of the analysis of snowpack
simulations with the original use of categorical scores to separate different physical processes.

The organization of the paper is as follows. In section 2, we introduce briefly the geographical and climate characteristics of the study area and period. Section 3 describes the snowpack model Crocus; then, the atmospheric forcing from NWP model AROME at kilometric resolution, and the forcing from SAFRAN reanalysis; finally, the observations dataset and verification methods. Sec-
tion 4 details the results following three main axes: (i) global scores and spatial distribution of snow depth; (ii) daily snow depth variations and winter precipitation; and (iii) comparison to Snow Water Equivalent scores and study of bulk snowpack density. These results are discussed in section 5, with concluding remarks and outlooks.



## 2 Study area and period

The study area is the Pyrenean chain (Fig. 1), extending from the Atlantic Ocean to the Mediterranean Sea, following the France/Spain border. Several summits exceed 3000 m.a.s.l. in the central part, reaching a maximum altitude of 3404 m.a.s.l. at Aneto Peak in Spain. Our domain of study covers France, Andorra and Spain, between latitudes 41.6°N and 43.6°N and longitudes -2.5°E and 3.5°E (approximately 500 km x 220 km).

The Pyrenean climate closely follows the oceanic influence. This region is predominantly exposed to westerly winds; the prevailing disturbances have an Atlantic origin and are weakened in the Eastern part. Thus, most of winter precipitations, controlling the snow cover distribution, are due to South-West to North-West flows (e.g. Buisan et al., 2015; Durand et al., 2012; Maris et al., 2009; Vada et al., 2013). It generates a strong West-East gradient of decreasing precipitation, leading to

a similar gradient of mean snow depth and number of days with snow on the ground (Maris et al., 2009). A North-South gradient of snow quantities (with more snow on the Northern side) is due to warmer and drier conditions in Spain than in France, largely associated to a frequent foehn effect in Spain by North flows (López-Moreno et al., 2009). According to Maris et al. (2009), we defined three climatic regions: Western Pyrenees, under the direct influence of the Atlantic Ocean, Central

Pyrenees, with a more continental climate, and Eastern Pyrenees, under the Mediterranean influence (Fig. 1).

The study period goes from August 2010 to July 2014. Because of the interannual variability of winter season conditions, several years are necessary to perform a significant evaluation of snow models (Essery et al., 2013). Moreover, the 2010/2014 period covers four very contrasted winters.

The conditions were rather dry during winter 2010/2011, which led to a deficit of snow in the Pyrenees (with respect to the climate normal), despite early snow in November. Winter 2011/2012 was also characterized by a deficit of snow, particularly marked in the Spanish Pyrenees, due to dry conditions (Vada et al., 2013; Gascoin et al., 2015). In contrast, winter 2012/2013 was very cold and wet, leading to 40-years records of snowfall and snow depth, particularly in the French Pyrenees.

Winter 2013/2014 was also characterized by much higher levels of snow than normal, due to a lot of precipitation, despite warmer conditions.

## 3 Data and methods

### 3.1 Snowpack model

Snowpack simulations were carried out using the detailed snow cover model Crocus (Brun et al.,

1992; Vionnet et al., 2012) coupled with the ISBA land surface model within the SURFEX (EXternalized SURFace) simulation platform (Masson et al., 2013). SURFEX/ISBA/Crocus models the evolution of the physical properties of the snowpack, its stratigraphy (with a user-defined maximum




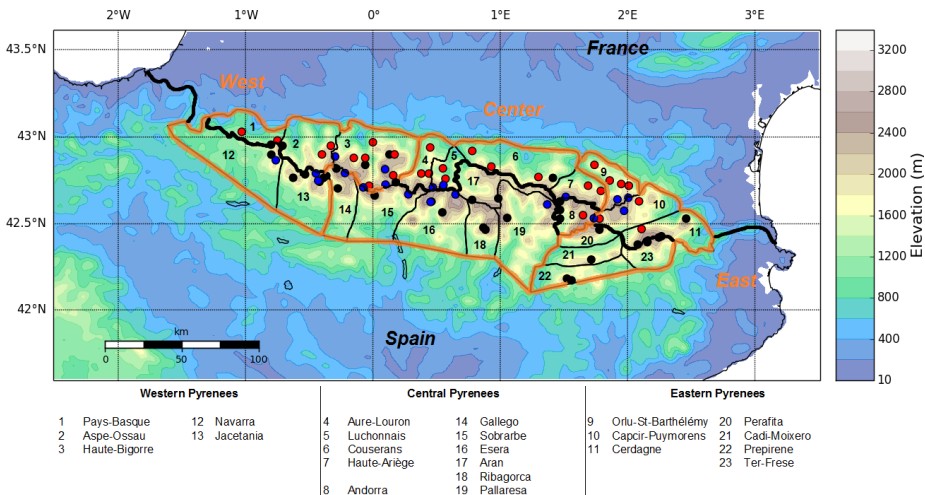

**Figure 1.** Location of measurement stations in the Pyrenees: SD and precipitation (red circles), SD and SWE (blue circles), SD only (black circles). Backgroung map: AROME topography (years 2010/2012). SAFRAN massifs delimited (black line), national borders (bold black line) and climatic regions (bold orange line). SAFRAN massifs names in caption.

number of layers, 50 in this study) and the underlying ground, under given meteorological forc-
ing data. The model is used here in offline mode (i.e. not fully coupled to atmospheric simulations),
with prescribed atmospheric forcing described in section 3.2. Snowpack simulations were performed
over the domain defined in section 2 (Fig. 1), on a regular 0.025°grid, from 1 August 2010 to 31 July
2014, with a 15 minutes internal time step.

Soil properties were obtained from HSWD 1 km resolution database for soil texture (FAO, 2012).
Aspect and slope are not taken into account for incoming solar radiations, since the 2.5 km resolution
topography can hardly represent the local orography of observation stations. As observations are
collected in open fields, the interactions with vegetation and the parameterization of fractional snow
cover are not activated within the SURFEX scheme. Wind-induced snow transport is not simulated.

### 3.2 Atmospheric forcing

Crocus requires the following atmospheric forcings: reference level temperature and specific humid-
ity (usually 2 m above ground), wind speed (usually 10 m above ground), incoming shortwave and
longwave radiations, solid and liquid precipitation. Two different forcings were used: one generated
from the AROME NWP system (Seity et al., 2011) operational forecasts and the other one from the
SAFRAN reanalyses (Durand et al., 1993, 2009b). These forcings are described hereafter.





### 3.2.1 AROME: kilometric resolution NWP system

AROME is the high resolution NWP system of Meteo-France (Seity et al., 2011). Its horizontal grid
mesh of 2.5 km (upgraded to 1.3 km in 2015, Brousseau et al., 2015) makes it of particular interest
for forecasting intense events (like convective rains) and small scale processes in alpine terrain, such
as orographic precipitations or foehn effects, thanks to a realistic description of the topography.
AROME is a spectral and non-hydrostatic model, combining the physical package of the research
model Meso-NH (Lafore et al., 1998) with the dynamical core of the non-hydrostatic version of the
limited area NWP ALADIN (Bubnová et al., 1995).

AROME physics schemes include the cloud microphysical scheme ICE3 (Pinty and Jabouille,
1998), with five prognostic hydrometeors (rain, snow, graupel, ice crystals and cloud droplets). The
radiations scheme is the parameterization of the European Centre for Medium-Range Weather Fore-
casts (ECMWF). The turbulence parameterization comes from the Turbulent Kinetic Energy scheme
of Cuxart et al. (2000), with the Bougeault-Lacarrere mixing length (Bougeault and Lacarrère, 1989).
The atmosphere vertical discretization is made of 60 levels with a lowest atmospheric level at 10 m
above ground. The SURFEX simulation platform (Masson et al., 2013) computes the energy, mass
and momentum exchanges, with the ISBA (Interaction between Soil Biosphere and Atmosphere)
scheme (Noilhan and Planton, 1989) over land, including two layers in the ground and a simple
one-layer snowpack scheme (Douville et al., 1995).

The implementation of AROME as an operational system is made through thirty-hours forecasts
at the 00:00, 06:00, 12:00, and 18:00 UTC nominal analysis times, over a domain covering France.
Data assimilation is based on a 3-hourly continuous cycle (Brousseau et al., 2008), generating 3D-
Var analysis with a 3-h assimilation window centered on the analysis time. We use here the hourly
forecasts issued from the 00:00 UTC analysis time, from +6h to +29h , extracted on a regular lat-
itude/longitude 0.025°grid to build a continuous forcing from 1 August 2010 to 31 July 2014 over
the studied domain.

Some changes in the operational configuration of AROME occured during the four years of sim-
ulations: the simulation domain was extended during summer 2012 with a modification of the to-
pographic database. Topography from Global 30 Arc-Second Elevation Data Set (GTOPO30) was
used in a low-resolution version (5 km) before summer 2012, and at 30 Arc-Second (approximately
1 km) resolution afterwards; leading to a change of the forcing files orography in the middle of our
simulation period.

### 3.2.2 SAFRAN: analysis system

180

The SAFRAN analysis system (Durand et al., 1993, 2009a, b) provides hourly atmospheric forcing
data for each of the 23 massifs of the Pyrenees (Fig. 1). Within each massif, the forcing is provided
by 300 m altitude steps. SAFRAN reanalyses take a preliminary guess from the global NWP model





ARPEGE (from Meteo France, 15 km grid spacing guess projected on a 40 km grid), complemented
by available observations from automatic weather stations, manual observations carried out in the
climatological network and in ski resorts and atmospheric upper-level sounding. In particular, a daily
precipitation analysis is included, with a climatological guess depending on a daily determination
of the general weather pattern. This determination is based on a classification of nine weather pat-
terns, defined by Meteo France mountain forecasters to be representative of the main precipitating
regimes of the Pyrenees. It is made following the synoptic circulation, through the altitude of the
$500\,\mathrm{hPa}$ geopotential level. In this study, SAFRAN forcing was interpolated over the $0.025°$ grid of
the domain described in section 2, following the method described by Vionnet et al. (2012).

### 3.3 Evaluation dataset: snowpack and precipitation measurements

The observational dataset contains snow depth (SD), snow water equivalent (SWE) and precipita-
tion measurements available in the Pyrenean SAFRAN massifs, both in France and Spain. The SD
observations consist of daily manual measurements at ski resorts (at 6 UTC) and hourly automatic
measurements by ultra-sonic sensors at high altitude stations; only the value at 6 UTC from the
hourly record is used in this study. The SWE measurements come from automatic stations with
cosmic ray snow gauges (Gottardi et al., 2013): daily values are obtained through a 24h-median
smoothing of hourly measurements. Both SD and SWE data are independent (i.e. not assimilated
in SAFRAN–Crocus or AROME–Crocus simulations). The 24h-cumulated precipitations measure-
ments are manually collected every day at ski resorts with precipitation gauges (at 6 UTC), without
any correction. These data ere assimilated in SAFRAN.

A criterium of altitude is then applied to select adequate stations: only stations with less than 150
meters elevation difference to the model topography are selected for evaluation. After this selection,
83 SD stations could be used in the whole Pyrenees, amongst which 20 stations including SWE
measurements and 28 stations with precipitation measurements (Fig. 2). 45 of them are located in
France, 38 in Spain, 24 in the Western Pyrenees, 32 in the Central Pyrenees and 17 in the Eastern
Pyrenees (Fig. 1). These stations are all between 1000 m.a.s.l. and 2600 m.a.s.l. The altitude dis-
tribution is represented on Fig. 2. The mean altitude, weighted by the number of SD observations,
is 2007 m.a.s.l. The spatial extent of stations on the domain can be considered representative (all
massifs have observations), except a lack of data in the Southern foothills that may be deplored.

### 3.4 Evaluation methods

AROME–Crocus snowpack simulations were evaluated in terms of SD and SWE from October 1
to June 30 over the period 2010/2014. SAFRAN–Crocus simulations were evaluated in a similar
manner. Three error metrics were used: the bias, the Root Mean Square Error (RMSE) and the
Standard Deviation Error (STDE, which represents the temporal and spatial dispersion around the
bias).





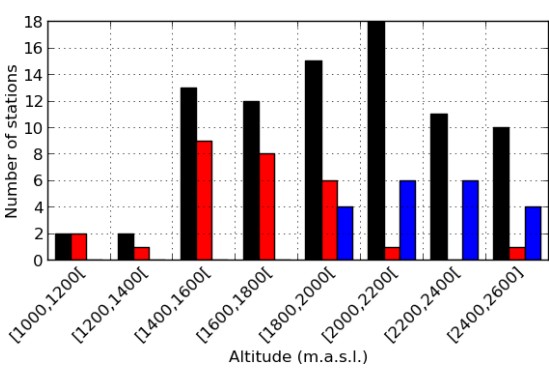

**Figure 2.** Altitude distribution of all SD stations (black), precipitation gauges (red) and SWE stations (blue).

**Table 1.** 2x2 contingency table

|  | OY | ON |
|---|---|---|
| FY | HI (hits) | FA (false alarms) |
| FN | MI (misses) | CR (correct rejections) |

OY = Observed Yes; ON = Observed No
FY = Forecast Yes; FN = Forecast No

A complementary evaluation is carried out in terms of daily snow depth variations. This additional metrics allows to avoid cumulative errors, occurring during a winter season, and to offer another view on precipitation forecast as well as the simulation of settling and ablation processes. The daily Snow Depth variation $\Delta SD_j$ is defined for day $j$ as:

$$\Delta SD_j = SD_j - SD_{j-1} \tag{1}$$

$\Delta SD$ categories are defined for decrease and accumulation, and allow to study categorical distribution, sums and scores, in a similar way as Schirmer and Jamieson (2015) for their study of winter precipitations. Daily Snow Water Equivalent variation ($\Delta SWE$) is also defined in the same way.

A categorical score was used to study daily variations: the Equitable Threat Score (ETS), according to its definition by Nurmi (2003), based on 2x2 contingency tables (Table 1). The ETS is a score commonly used for precipitation forecast evaluation (e.g. Bélair et al., 2009). It was used here for comparison to Schirmer and Jamieson (2015). It measures the proportion of correct "yes"-events amongst all events, except correct rejections (the forecast skill does not consider "no"-events, much more frequent than "yes"-events):

$$ETS = \frac{HI - HI_{rdm}}{HI + FA + MI - HI_{rdm}} \tag{2}$$

and taking into account hits by chance:

$$HI_{rdm} = \frac{(HI + FA)(HI + MI)}{N} \tag{3}$$





**Table 2.** Scores (bias and STDE) for simulated snow depth against observations in the Pyrenees for winters 2010/2011 to 2013/2014

| | stations | N | mean obs. (cm) | bias (cm) | | STDE (cm) | |
| --- | --- | --- | --- | --- | --- | --- | --- |
| | | | | AROME | SAFRAN | AROME | SAFRAN |
| 2010-2014 | 83 | 47169 | 70 | 55 | 22 | 70 | 57 |
| 2010-2011 | 63 | 10445 | 48 | 57 | 20 | 55 | 42 |
| 2011-2012 | 62 | 10401 | 39 | 43 | 16 | 52 | 44 |
| 2012-2013 | 79 | 14281 | 103 | 52 | 17 | 77 | 65 |
| 2013-2014 | 67 | 12042 | 76 | 65 | 37 | 85 | 64 |
| West | 27 | 14393 | 83 | 65 | 17 | 84 | 54 |
| Center | 35 | 21865 | 72 | 57 | 28 | 64 | 55 |
| East | 21 | 10911 | 50 | 36 | 18 | 58 | 63 |
| France | 45 | 22491 | 76 | 56 | 17 | 75 | 50 |
| Spain | 38 | 24678 | 65 | 53 | 28 | 66 | 62 |
| [1000m,1800m[ | 29 | 11975 | 48 | 66 | 25 | 71 | 43 |
| [1800m,2200m[ | 33 | 19164 | 76 | 46 | 17 | 72 | 61 |
| [2200m,2600m[ | 21 | 16030 | 80 | 57 | 27 | 66 | 61 |

where $N = HI + FA + MI + CR$ is the total number of observations. It ranges from -1/3 to 1, where 0 means no skill and 1 means perfect score.

Evaluations of AROME and SAFRAN atmospheric forcing were also conducted in the same period in terms of daily precipitation. The Heidke Skill Score (HSS) was used for verification, with persistence (observation of the day before) as reference:

$$HSS = \frac{(HI - HI_{per}) + (CR - CR_{per})}{N - HI_{per} - CR_{per}} \qquad (4)$$

The HSS allows to compare the skill of two models. It is a standard skill score used for precipitation forecast evaluation of NWP systems(Amodei and Stein, 2009). It ranges from $-\infty$ to 1, where 0 means no skill and 1 means perfect score.

## 4   Results

### 4.1   Evaluation of simulated snow depth

#### 4.1.1   Global scores for the winter season

Table 2 summarizes error statistics for snow depth during the whole period of study. The number of stations available varies every year (from 62 to 79) because of the change in model topography





and missing data. Scores have also been computed for a constant number of stations (restricted to 46, not shown), and showed that the annual variability of the number of stations does not impact the results and analysis exposed hereafter. These scores show a global overestimation of snow depth by AROME–Crocus with an overall bias of + 55 cm, while the overall bias of SAFRAN–Crocus is + 22 cm. The overall STDE reaches 70 cm for AROME–Crocus, against 57 cm for SAFRAN–Crocus.

The errors are rather high for both models; some elements of explanation will be given in the next sections.

For both models, highest STDEs are found in winters 2012/2013 and 2013/2014, two very snowy winters. In terms of spatial distribution, the positive bias and STDE decrease from West to East for AROME–Crocus, with notable errors in the Western zone. In the Eastern zone, AROME–Crocus

and SAFRAN–Crocus STDEs are equivalent. AROME–Crocus scores are equivalent in France and Spain, while SAFRAN–Crocus behaves slightly better in France, probably due to a higher number of observations assimilated in the model. In terms of altitudinal range, biases are constant for SAFRAN–Crocus and decrease for AROME–Crocus, which implies an higher relative bias in the [1000 m, 1800 m[ range.

Figure 3 shows scores for each station on the whole period of study. Almost all stations show an overestimation of snow depth, particularly for AROME–Crocus with extreme positive biases on the Atlantic foothills. The 3 highest biases for AROME–Crocus are at the 3 stations around Pic d'Anie, first summit above 2500 m.a.s.l. close to the Ocean : Isaba El Ferial (+ 188 cm; massif of Navarra, Western Pyrenees, Spain), Arette La Pierre Saint Martin (+ 209 cm; massif of Pays-

Basque, Western Pyrenees, France), Soum Couy Nivôse (+ 229 cm; massif of Aspe-Ossau, Western Pyrenees, France). These 3 stations also show a very high STDE (higher than 1 m). The 2 next highest biases are located in the North-West foothills: Gourette (+ 135 cm; massif of Aspe-Ossau, Western Pyrenees, France) and Hautacam (+ 154 cm; massif of Haute-Bigorre, Western Pyrenees, France). This region is particularly exposed to W to NW flows due to its proximity to the Ocean.

There is thus an excessive orographic blocking on these first reliefs by AROME. Except for these stations, biases and STDEs are more homogeneous in the rest of the Pyrenees.

### 4.1.2 Focus on winter 2011/2012

Winter 2011/2012 was characterized by a deficient snowpack in the Spanish Pyrenees, due to dry and warm weather in the Southern side of the chain (Vada et al., 2013). It was also characterized

by a strong contrast between the French and the Spanish sides of the Pyrenees: even if the French Pyrenees exhibited a deficit of snow most of the winter (with respect to the climate normal), the first half of February 2012 was exceptionally cold and snowy in France. Due to Northern flows, these snowfalls did not affect (or at a much lower magnitude) the Spanish side. This asymmetry (and the resulting drop in Spanish hydropower production in springtime) was highlighted in terms of snow

cover duration in Gascoin et al. (2015). We show here the added value of AROME high-resolution





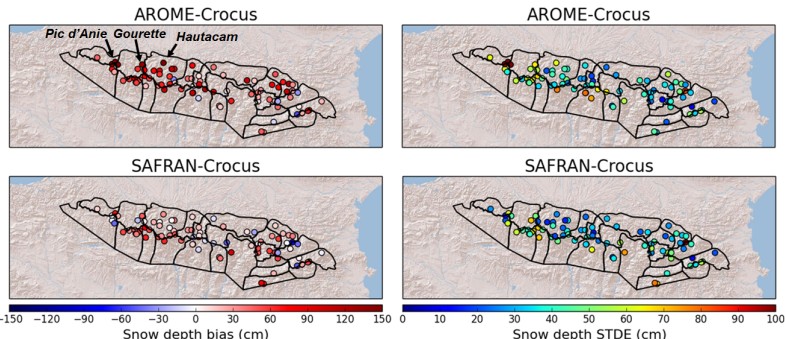

**Figure 3.** Snow depth bias (left) and STDE (right) by station for AROME–Crocus (up) and SAFRAN–Crocus (down), 2010/2014

forcing for simulating a particular meteorological contrast due to the topography, and the resulting snow cover distribution.

Figure 4 gives an overview of the snow cover simulated by AROME–Crocus and SAFRAN–Crocus (values of SWE higher than 20 mm), compared to MODIS fractional snow cover images (MOD10A1, Klein and Stroeve, 2002), on the 22 February 2012. This date (selected because of clear sky conditions) is close to the end of the intense cold and snowy events of the first half of February in French Pyrenees, corresponding to a maximum contrast between both sides of the Pyrenees. This contrast appears clearly on MODIS snow cover image, where snow is only present on the highest summits of the Spanish Pyrenees, on the border ridge; while the snow covers most of the French Pyrenean massifs and Val d'Aran (in Spain, but in the Northern side of the Pyrenean highest ridge). The absence of snow in Spanish Pyrenean foothills is particularly well represented in the AROME–Crocus simulation, and the snow cover distribution is notably faithful to the observations. On the contrary, SAFRAN–Crocus simulation exhibits a rather homogeneous snow cover in Spanish massifs (despite still lower quantities than in the French Pyrenees). The snow cover spatial distribution, and particularly the snow deficit in the Spanish Pyrenees, is thus better simulated by AROME–Crocus.

This improvement in terms of snow cover may be attributed to AROME dynamical behaviour in complex topographies. Vada et al. (2013) showed that the snowfall deficit in 2011/2012 was more sensitive at Spanish stations exposed to South flows, while Spanish stations more exposed to North flows exhibited a lower negative anomaly. The snowpack was mainly constituted by N-NW flows during this season, which is confirmed by a study of SAFRAN weather patterns. We cumulated all snowfalls (from SAFRAN outputs) occuring on the studied domain between 1 October 2011 and 22 February 2012 (date studied on Fig. 4). 67% of total snow quantities fell during days of North to North-West flows, which correspond to two synoptic patterns: minimum geopotential on Genova gulf and maximum on Ireland, associated to N and NW flows (38%); and disturbed NW flow with strong geopotential gradient, implying strong precipitations on the NW French Pyrenees and foehn





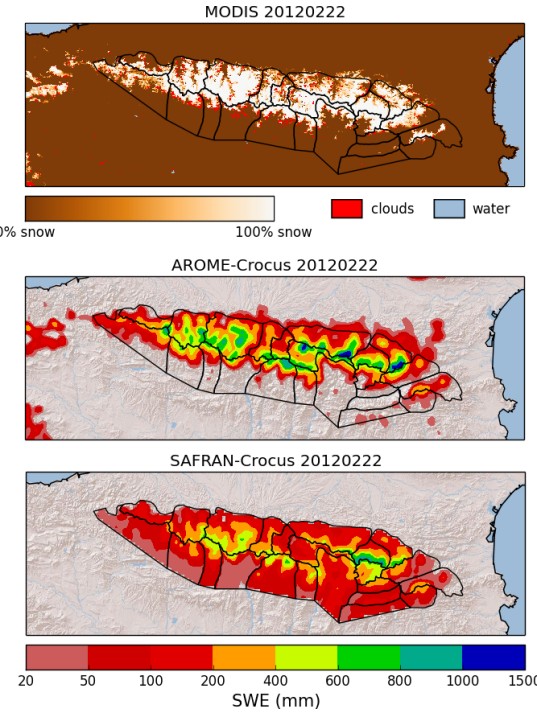

**Figure 4.** Top: snow cover fraction on 22 February 2012, from MO10A1 images. Bottom: SWE simulations by AROME–Crocus and SAFRAN–Crocus, same date. SAFRAN–Crocus simulations are only defined within SAFRAN massifs.

effect in Spain (29%). On the four winters 2010/2014, these synoptic conditions constitute 45% of total snowfalls. In contrast, only 4% of total snow quantities fell during days of South to South-West flows (against 14% on period 2010/2014).

The behaviour of both forcing models in such specific synoptic conditions is of particular interest.

315 Snowfalls from AROME and SAFRAN were cumulated from 1 October 2011 to 22 February 2012. They are represented on Fig. 5 along a NW/SE cross section. Orographic blocking is visible on the windward sides, with maximum snowfall shortly upstream from the highest summit; and foehn effect in Spain implies a drastic drop of snowfalls immediately behind the highest ridge. The orographic shield of the first high mountains of the Haute-Bigorre massif implies snowfall weaker than upstream

320 at the same altitude (approximately four times less). This windward/leeward distinction within a massif is not simulated by SAFRAN, since two points at the same altitude and within the same massif get the same amount of snowfall. The difference between both forcings is marked in Esera (Spanish massif), where the orographic shield and resulting dry weather is not enough represented by SAFRAN, compared to AROME. Such differences are even more marked when filtering only





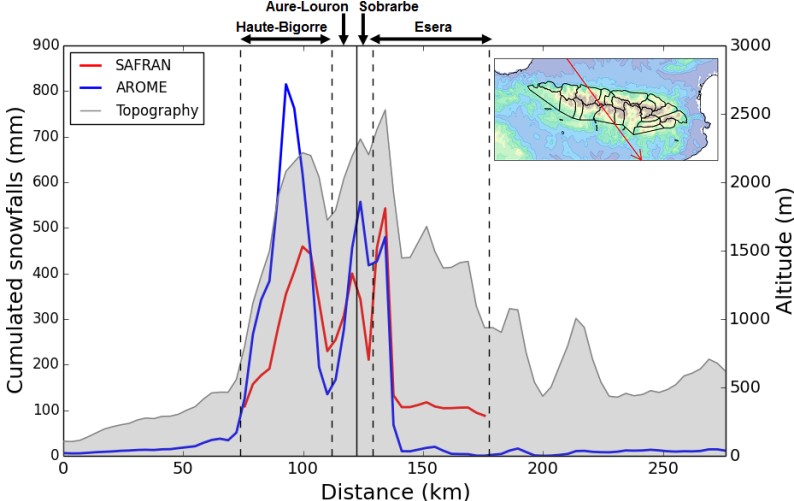

**Figure 5.** Cross section of cumulated snowfall from 1 October 2011 to 22 February 2012 for AROME forecasts (blue) and SAFRAN reanalysis (red), with topography plotted on the right axis in grey. The location of the transect is given on the upper right map.

cumulated snowfalls occuring by N-NW flows (not shown). This study emphasizes the added value of AROME dynamics, which allow to better take into account mesoscale orographic effects.

### 4.2 Daily SD variations

#### 4.2.1 Global scores

The RMSE of daily $\Delta SD$ indicates the ability of the model to forecast (or analyse) the appropriate evolution of snow depth. This score was computed for AROME–Crocus and SAFRAN–Crocus. It is equal to 7 cm for both models, with low spatial variation. The RMSE is slightly higher in the most snowy winters (8 cm in 2012/2013 and 2013/2014 against 6 cm in 2010/2011 and 2011/2012). This is a first complementary information to global scores that indicate that, despite an overall overestimation, AROME–Crocus gives similar results compared to SAFRAN–Crocus in terms of daily snow depth variations.

#### 4.2.2 Categorical scores

A categorical study of $\Delta SD$ enables to understand the models behaviour in terms of SD increase (accumulation) and decrease (ablation and settling). The categorical frequency distribution of $\Delta SD$ is plotted on Fig. 6, according to eight accumulation categories, two decrease categories and one "no variation" category [-0.2 cm, 0.2 cm[. Small daily accumulations (between 0.2 cm and 10 cm per day) are overrepresented by both models, while the occurrence of medium and high daily ac-





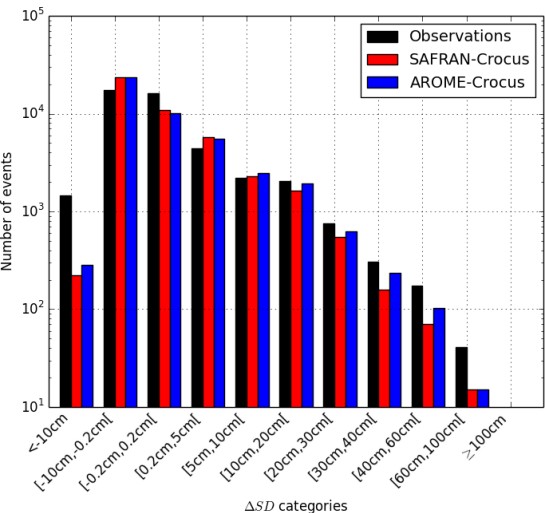

**Figure 6.** Categorical frequency distribution of $\Delta SD$ for observations (black), AROME–Crocus (blue) and SAFRAN–Crocus (red), at all stations, 2010/2014.

cumulations (more than 10 cm per day) is underestimated by both models. However, the frequency of medium and high accumulation events forecast by AROME–Crocus is systematically closer to the observations than SAFRAN–Crocus. There is also a clear discrepancy between models and ob-

servations for the strong decrease category, largely underestimated by both AROME–Crocus and SAFRAN–Crocus.

In terms of quantities, categorical sums of $\Delta SD$ are plotted on Fig. 7. SAFRAN–Crocus strongly underestimates the high accumulation quantities. AROME–Crocus is closer to observations for these categories (particularly for the [10 cm, 20 cm[ category, main contribution to the snow accumula-

tion). It is mechanically counterbalanced by an overestimation of small accumulation quantities. The sum of all accumulation categories shows an overall underestimation of snow accumulation by both models: the total sum of observed accumulations is 904 m, against 857 m for AROME–Crocus (- 5 %), and 753 m for SAFRAN–Crocus (- 17 %). The largest gap concerns the category of strong decrease, globally missed by models. Since AROME–Crocus and SAFRAN–Crocus un-

derestimate accumulations, this category is the main contributor to the overall overestimation of snow depth: the positive bias shown in section 4.1.1 is not due to excessive snowfall but to unsufficient snow depth decrease. Total decrease quantities are more pronounced for AROME–Crocus than SAFRAN–Crocus as a logical consequence of more marked accumulations. Plotting the cumulated $\Delta SD$ by altitudinal range (under 1800 m, between 1800 m and 2200 m, and above 2200 m) high-

lights a similar behaviour of models, except a stronger underestimation of high accumulations by SAFRAN–Crocus at lowest altitudes (not shown).





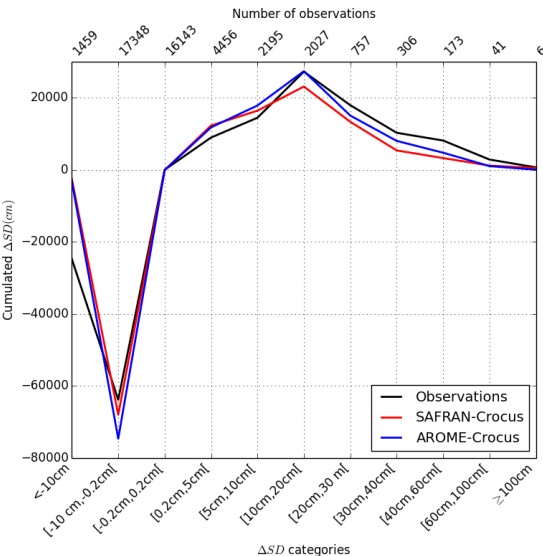

**Figure 7.** Cumulated $\Delta SD$ by categories for observations (black), AROME–Crocus (blue) and SAFRAN–Crocus (red), at all stations, 2010/2014.

In order to isolate the specific behaviour of AROME–Crocus in the Atlantic foothills, $\Delta SD$ categorical distribution is plotted on Fig. 8 for the three stations near Pic d'Anie, where the positive bias was found to be the highest in section 4.1.1. In contrast to its general behaviour, AROME–Crocus

strongly overestimates accumulations, particularly strong accumulations. At the same time, strong decreases are also underestimated, which results in a rather high positive bias.

### 4.2.3 Study of accumulation processes and comparison to precipitations

The performance of models for daily snow accumulations is further studied thanks to the ETS, computed for threshold categories (Fig. 9). Scores are similar for AROME–Crocus and SAFRAN–

Crocus. The ETS is almost 0.40 for the "all accumulations" category (more than 0.2 cm) and is under 0.10 for high accumulations (more than 40 cm). SAFRAN–Crocus has a better ETS for small accumulations, but the ETS of AROME–Crocus is better for all accumulations higher than 10 cm, except extreme accumulations (more than 60 cm). However, the very small sample size for this category (47 observed events) makes impossible any reliable interpretation. A distinction by altitudinal range

shows equivalent ETS for AROME–Crocus and SAFRAN–Crocus above 1800 m, and higher ETS for AROME–Crocus for medium and strong accumulations under 1800 m (not shown).

A complementary information on winter precipitation comes from the network of gauges in the French Pyrenees (red dots on Fig. 1). Daily accumulations of precipitation (rainfall plus snowfall, cumulated from 6UTC to 6UTC) from the forcing models are then directly compared to precipi-





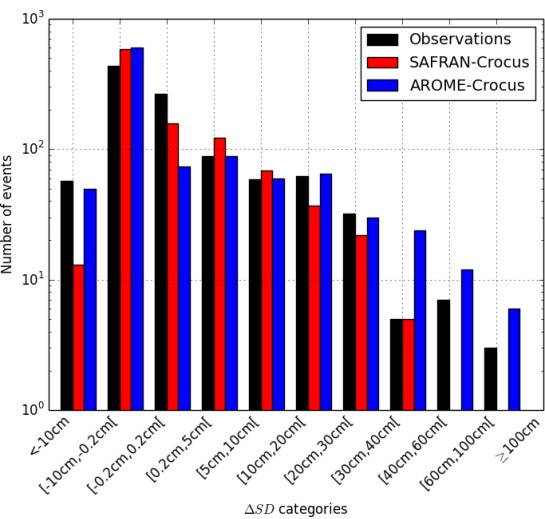

**Figure 8.** Categorical frequency distribution of $\Delta SD$ for observations (black), AROME–Crocus (blue) and SAFRAN–Crocus (red), at three stations near Pic d'Anie, 2010/2014.

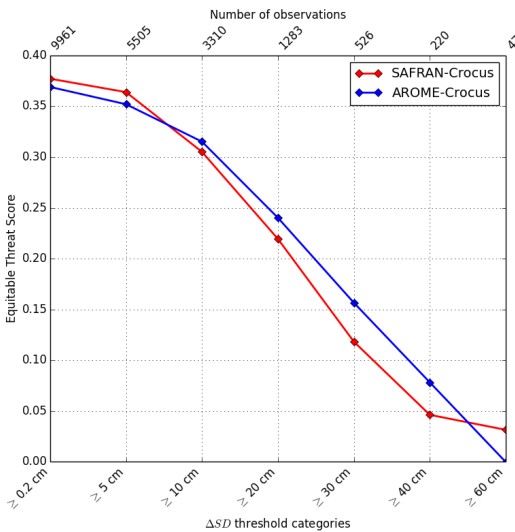

**Figure 9.** ETS of $\Delta SD$ threshold categories for AROME–Crocus (blue) and SAFRAN–Crocus (red), 2010-2014.

tation gauges measurements, for months from December to March (DJFM) in order to reduce the proportion of rainfall amongst precipitation. Most of these observations are assimilated in SAFRAN reanalyses, while they are not taken into account in AROME forecasts. Figure 10 represents the





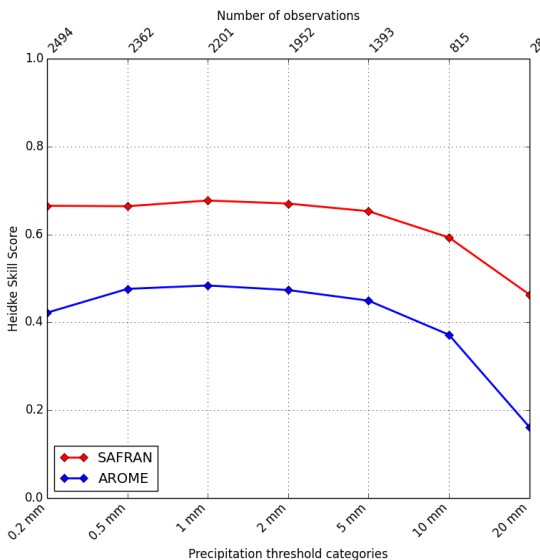

**Figure 10.** HSS of precipitation threshold categories for AROME–Crocus (blue) and SAFRAN–Crocus (red), period DJFM, 2010-2014.

HSS of both models for threshold categories, relatively to persistence. SAFRAN HSS is clearly higher than AROME HSS (between +0.18 and +0.30), due to the assimilation of the observations in
SAFRAN reanalyses. AROME HSS ranges from 0.16 for 20 mm threshold to 0.48 for 1 mm threshold. These scores are slightly lower than those reported for AROME in the Alps by Vionnet et al. (2015b). This lower performance compared to the Alps is also found in SAFRAN scores (Vionnet et al., 2015b).

Figure 11 shows cumulated precipitations by category for both models and observations (right)
compared to cumulated $\Delta SD$ at the same stations (left): in contrast to $\Delta SD$, AROME overestimates precipitations measured by gauges (+ 46 %). The optimal interpolation basis of the SAFRAN analysis system should mathematically not be biased on the assimilated observations over a long period. The slightly positive bias obtained in this study (+ 14 %) may be linked to the fact that some assimilated observations are not included in our evaluation dataset and/or to differences be-
tween the climatological guess and the mean precipitation amount of the 4 studied years. The strong overestimation of AROME is particularly notable for the largest amounts (+ 173 % for the > 50 mm category). The different distribution of precipitation and $\Delta SD$ for AROME, with a higher proportion of strong precipitation than strong snow accumulations, may be due to the fact that the precipitation considered here include rainfall and snowfall: amongst AROME total precipitation on the figure, 25
% of them are rainfalls, which contribute to settling and melting. Moreover, the stronger the snowfall, the stronger the snowpack settles under its own mass, which shifts the distribution to the left.





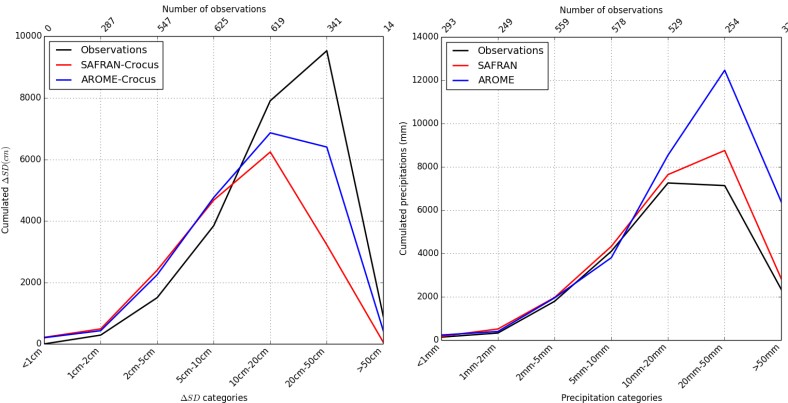

**Figure 11.** Cumulated $\Delta SD$ (left) and precipitation (right), for observations (black), AROME–Crocus (blue) and SAFRAN–Crocus (red), by categories, at the 28 same stations with SD and precipitation measurements, period DJFM, 2010-2014.

The overestimation of precipitation by AROME compared to precipitation gauges seems to be an apparent paradox, as we highlighted an opposite behaviour in term of snow accumulation. This theoretical discrepancy can be explained by the quality of precipitation gauges measurements. The
undercatch of solid precipitations by gauges, mainly due to wind effects, is well known and very variable: this issue is tackled by the WMO Solid Precipitation InterComparison Experiment (e.g. Wolff et al., 2015). There is no undercatch correction applied to these manual measurements, which implies that real precipitation amounts can be underestimated in the observations under windy conditions.

**4.2.4 Study of ablation processes**

A major part of model positive bias in SD is due to the underprediction of strong SD decreases. Consequently, the understanding of models biases implies a more developed study of ablation processes. Strong decreases larger than 10 $\mathrm{cm.day}^{-1}$ can be related to ablation processes like melting or wind-induced erosion, that need to be studied independently. To this end, two diagnostics have
been applied to identify such events. Melting snow days (MSD) correspond to days when snow upper layer temperature is equal to the melting point at 12UTC, in SAFRAN–Crocus outputs (no snow surface temperature measurement available). Wind-blown snow days (BSD) are identified at automatic weather stations only, where 10m-wind measurements are available. BSD correspond to days when 10m-wind speed exceeds 8 $\mathrm{m.s}^{-1}$ during more than 10 minutes and no melting is diagnosed
(only dry snow can be drifted). This value is based on the estimate of wind threshold for dry snow transport by Li and Pomeroy (1997). These criteria are obviously quite rough, but the visual check of correspondence with snow depth plots is satisfying. As illustration, the diagnosed days are reported





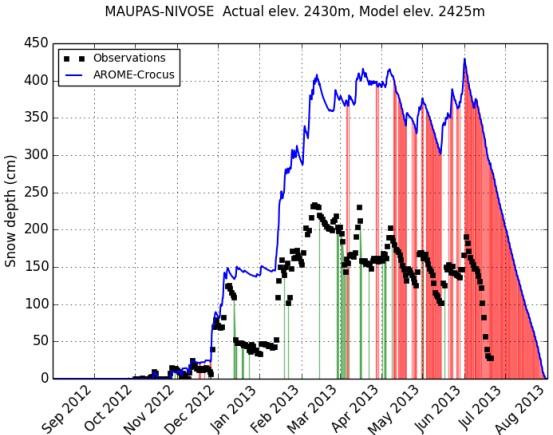

**Figure 12.** Snow depth simulated by AROME–Crocus (blue line) and observed (black squares) at Maupas station, 2012/2013. Wind-blown snow days are identified in green and melting snow days in red.

on Fig. 12 together with the snow depth evolution measured and simulated by AROME–Crocus, at Maupas automatic station (massif of Luchonnais, Central Pyrenees, France), where blowing snow

events are known to be frequent. For instance, a 60 cm snow depth drop occurs during one day in mid-December 2012: it corresponds to a BSD diagnostic. MSD are mainly spotted after April 2013 and correspond to decreasing snow depth.

To quantify the impact of wind-blown snow events on models performance, the cumulated $\Delta SD$ for AROME–Crocus and observations are plotted on Fig. 13, including or not BSD, with a finer

categorization of SD decreases. All decreases are reduced for observations, in the strongest proportion for high decreasing rates (less than -20 cm), which means that wind-blown snow constitutes the main contribution to this category. For AROME–Crocus, BSD do not contribute to the strong ablation categories but to small ablation and accumulation categories in equivalent proportions.

Similarly, the cumulated $\Delta SD$ is plotted on Fig. 14 for only MSD. Very strong melting (more

than 20 $\mathrm{cm.day}^{-1}$) happens sometimes in observations, but never in simulations. Strong melting (between -20 $\mathrm{cm.day}^{-1}$ and -10 $\mathrm{cm.day}^{-1}$) is very underrepresented by models, while melting of less than 10 $\mathrm{cm.day}^{-1}$ is overrepresented.

Consequently, the underestimation of strong decreasing rates comes mainly from ablation processes: on the one hand, from wind-blown snow events which are not represented by models, as

they are small scale processes; and on the other hand, from an underestimation of strong snowpack melting (more than 10 $\mathrm{cm.day}^{-1}$).





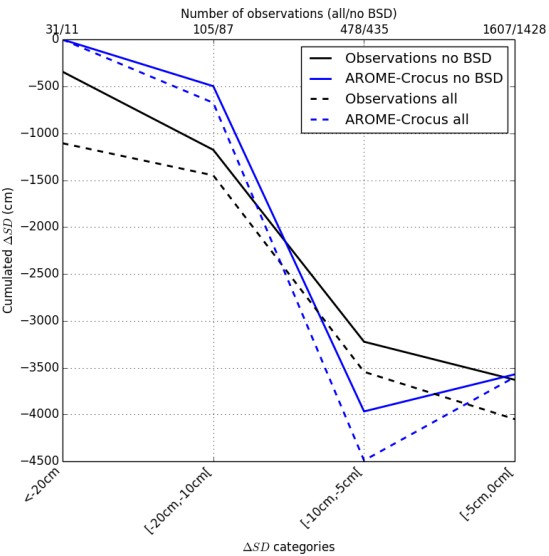

**Figure 13.** Cumulated $\Delta SD$ for AROME–Crocus (blue) and observations (black) by categories at seven high altitude stations, including wind-blown snow days (dashed lines) or not (solid lines), 2010-2014.

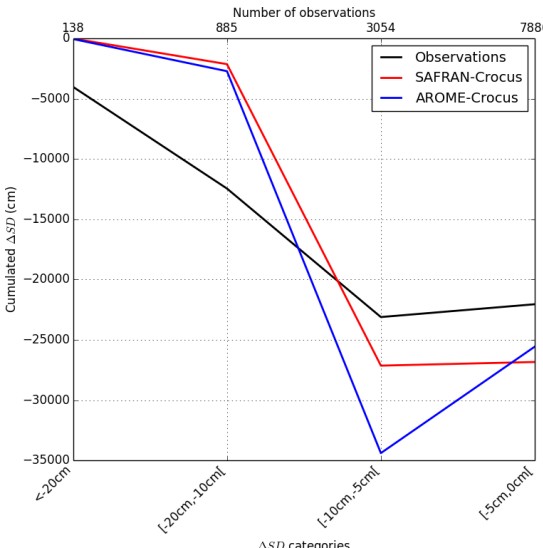

**Figure 14.** Cumulated $\Delta SD$ for AROME–Crocus (blue) and observations (black) by categories at all stations, melting snow days only, 2010-2014.



**Table 3.** Scores for simulated SWE and SD against observations in 20 high-altitude automatic stations in the Pyrenees for winters 2010/2011 to 2012/2013

| SWE | stations | N | mean obs. (mm) | bias (mm) | | STDE (mm) | |
|---|---|---|---|---|---|---|---|
| | | | | AROME | SAFRAN | AROME | SAFRAN |
| 2010-2013 | 20 | 14575 | 378 | 124 | -35 | 272 | 277 |
| 2010-2011 | 20 | 4979 | 282 | 139 | -5 | 208 | 179 |
| 2011-2012 | 20 | 4877 | 248 | 134 | 26 | 212 | 219 |
| 2012-2013 | 19 | 4719 | 614 | 96 | -130 | 367 | 375 |
| SD | stations | N | mean obs. (cm) | bias (cm) | | STDE (cm) | |
| | | | | AROME | SAFRAN | AROME | SAFRAN |
| 2010-2013 | 19 | 13111 | 92 | 50 | 10 | 61 | 57 |
| 2010-2011 | 19 | 4405 | 74 | 53 | 12 | 50 | 41 |
| 2011-2012 | 17 | 4222 | 57 | 55 | 20 | 57 | 54 |
| 2012-2013 | 19 | 4484 | 142 | 41 | -3 | 73 | 69 |

### 4.3 Snow Water Equivalent and bulk snowpack density

20 stations in the Pyrenees also include SWE measurements from 2010/2011 to 2012/2013: Table 3 summarizes the scores (bias and STDE) for SWE (upper part of the table). These stations are mainly above 2000 m.a.s.l (Fig. 2) and, thus, are not a representative sample of all SD stations of the Pyrenees. Consequently, SD scores at the same stations are added in the lower part of Table 3 for an adequate comparison. While SD scores follow the tendency indicated previously (strong overestimation for AROME–Crocus, slighter overestimation for SAFRAN–Crocus), SWE scores show a lower overestimation by AROME–Crocus in relative values (+ 33 % for SWE, + 54 % for SD, period 2010/2013) and a slight underestimation by SAFRAN–Crocus ( - 9 % for SWE, against + 10 % for SD). The STDE is equivalent between both simulations, even slightly lower for AROME–Crocus.

A categorical study of daily $\Delta SWE$ confirms the results exposed previously in terms of $\Delta SD$: the categorical sums of $\Delta SWE$ (Fig. 15) shows similarly an underestimation of strong accumulations and ablations by AROME–Crocus. In the same way, quantities simulated by AROME–Crocus are closer to observations than SAFRAN–Crocus. Cumulating all positive $\Delta SWE$ gives an estimate of total winter precipitations: the total sum of observed accumulations is 118.6 m, against 82.8 m for AROME–Crocus (- 30 %), and 61.7 m for SAFRAN–Crocus (- 48 %). The hypothesis of underestimation of snow accumulations mentioned previously is here confirmed, with a much stronger magnitude in terms of SWE than SD. This difference of magnitude goes together with the difference





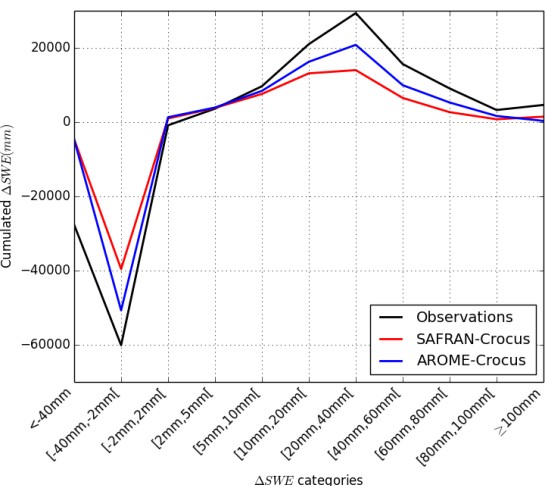

**Figure 15.** Cumulated $\Delta SWE$ by categories for observations (black), AROME–Crocus (blue) and SAFRAN–Crocus (red), 20 stations, 2010/2013.

of relative bias, regarding SWE or SD, reported in Table 3. In addition to accumulation and ablation processes, a complementary study is necessary to explain this discrepancy.

It suggests to investigate further bulk snowpack density in the simulations. SWE and SD automatic measurements in the 20 stations are made at the same point, which enables to compute a bulk snow-

pack density: $\rho = SWE/SD$ with $\rho$ in kg.m$^{-3}$, $SWE$ in kg.m$^{-2}$ and $SD$ in m. As SWE and SD measurements do not cover exactly the same surface footprint, we only consider snowpacks deeper than 20 cm to avoid problems of local heterogeneity, e.g. due to patchy snow cover in melting season. AROME–Crocus and SAFRAN–Crocus both have a negative bias of - 50 kg.m$^{-3}$ for a mean observation of 382 kg.m$^{-3}$: the bulk snowpack density is mainly driven by the snowpack model,

even if meteorological conditions are also involved. Consequently, the bias in terms of SD is necessarily higher than the bias in terms of SWE: a good simulation of SWE will lead to an overestimation of SD because of a too low bulk snowpack density. Fig. 16 shows the mean and standard deviation of simulated and observed $\rho$, at the 20 stations, for periods of 10 days, during winters 2011/2012 (left) and 2012/2013 (right). Both winters have very different snow cover evolutions. As exposed

previously, winter 2011/2012 is characterized by a rather thin snowpack, which implies a strong variability of $\rho$ and high bulk densities during all winter. For instance, a 50 cm snowfall on bare ground occured in the beginning of November 2011; there was no other significant snowfall during the month of November, with mild temperatures. That caused a quick settling, often associated with melting, and so a strong densification of the thin snowpack until the beginning of December (mean

observed $\rho$ of 450 kg.m$^{-3}$). Winter 2012/2013 was very cold and wet (Vada et al., 2013), with a very deep snowpack: a rather continuous densification of the snowpack occurs during the whole season.





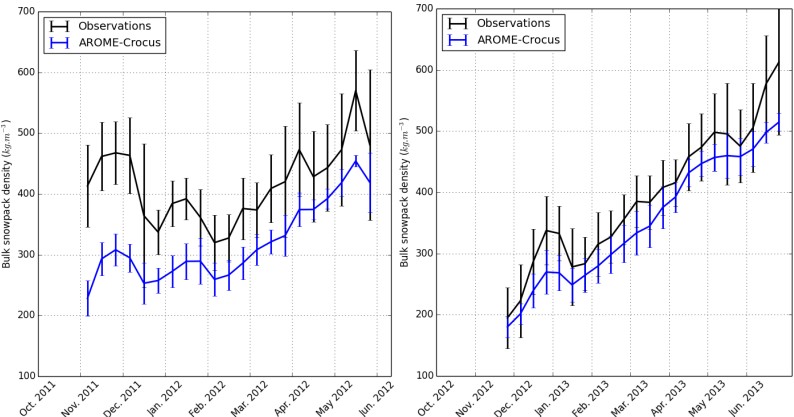

**Figure 16.** Bulk snowpack density during winters 2011/2012 (left) and 2012/2013 (right), mean of AROME–Crocus simulation (blue) and observations (black), at 20 stations, for periods of 10 days. Errorbars represent standard deviation.

The negative bias of AROME–Crocus is stronger for winter 2011/2012 (- 88 $\mathrm{kg.m^{-3}}$ for a mean observation of 403 $\mathrm{kg.m^{-3}}$, thin and dense snowpack) than winter 2012/2013 (-37 $\mathrm{kg.m^{-3}}$ for a mean observation of 385 $\mathrm{kg.m^{-3}}$, deep and less dense snowpack). Both snowpacks reach 550 to 600 $\mathrm{kg.m^{-3}}$ (firn density) at the very end of the Spring (end of May in 2012 and end of June in 2013).

A typical example of seasonal evolution of the bulk snow density is represented on Fig. 17, at station Les Songes (massif of Orlu, Eastern Pyrenees, France), during winter 2012/2013. $\rho$ is underestimated by AROME–Crocus during the whole season, particularly after long settling periods. Indeed, the densification slope is too low during the settling following a snowfall (increasing $\rho$, red arrows on Fig. 17). This is observable after every snowfall (decreasing $\rho$, green arrows on Fig. 17). For instance, fresh snow falls in the beginning of December 2012, with an adequate simulation of $\rho$ until then; the process of settling and densification of the snowpack occurs during the whole month of December reaching 350 $\mathrm{kg.m^{-3}}$ in observations, while the densification slope is much lower in simulations, reaching less than 300 $\mathrm{kg.m^{-3}}$.

## 5   Discussion and conclusion

A more accurate description of snow cover variability in mountainous terrain is necessary for many applications including mountain hydrology or avalanche hazard forecasting. In this paper, we have adressed the potential of the kilometer-scale NWP model AROME used as atmospheric forcing for distributed snowpack simulations in the Pyrenees. The simulations were carried out with the snowpack model Crocus at 2.5 km grid spacing, during four contrasted winters, from August 2010 to August 2014. They were evaluated through a comparison to simulations driven by the analysis





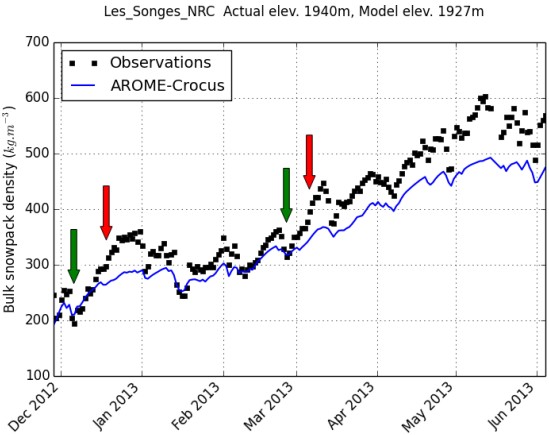

**Figure 17.** Bulk snowpack density observed (black) and simulated by AROME–Crocus (blue) at station Les Songes, winter 2012/2013. Green arrows indicate two examples of snowfalls, red arrows two examples of settling period.

system SAFRAN and to ground-based measurements of snow depth, snow water equivalent and precipitation across the whole mountainous chain. A global verification of Snow Depth simulation with 83 stations exhibited an overestimation in both simulations, with a higher positive bias for AROME–

Crocus than SAFRAN–Crocus. In terms of SWE (20 stations), the overestimation was less marked for AROME–Crocus and turned out to be an underestimation for SAFRAN–Crocus. In comparison to the evaluation performed by Vionnet et al. (2015b) in the French Alps, the overestimation is stronger in the Pyrenees (+ 55 cm against + 40 cm in the Alps for AROME–Crocus), but this difference is also present for SAFRAN–Crocus (+ 22 cm against + 17 cm in the Alps), which would

tend to ascribe its origin to the climatology (immediate vicinity and influence of the Atlantic Ocean and the Mediterranean Sea). However, for a longer time period, SAFRAN–Crocus does not exhibit such a bias over the French Pyrenees (Lafaysse et al., 2013), and the results may be specific to the studied seasons. The lowest biases were found in the Eastern part of the Pyrenees, which is also the driest; similarly to Vionnet et al. (2015b) who highlighted a lower overestimation in the Southern

Alps. The highest biases were found in the Western Pyrenees, which receive firstly and in highest quantities the precipitations coming from the Ocean.

The added value of AROME–Crocus to represent the spatial variability of the snowpack within each massif was emphasized on winter 2011/2012. AROME captures mesoscale orographic effects (enhanced precipitation on the upwind side of mountains, as shown on Fig. 5); thus enabling a

more adequate distribution of the snow cover compared to SAFRAN–Crocus. Vionnet et al. (2015b) showed this high variability within Alpine massifs in terms of seasonal snowfall. The dynamical behaviour of AROME, compared to SAFRAN, is of particular interest in a relatively narrow chain





such as the Pyrenees, where orographic blocking and foehn effects are very frequent, creating strong climatic and snowpack heterogeneities. Nevertheless, the orographic blocking was shown to be ex-
cessive in the first mountainous areas near the Atlantic Ocean, probably due to an excessive vertical updraft of the disturbed oceanic flows on the first strong slopes, or to an excessive model reactivity to these updrafts.

The study of daily SD and SWE variations enables a more detailed understanding of the scores of models. We indeed show that the global overestimation of SD and SWE is not the consequence of
overestimated snowfall (except in the Atlantic foothills). Snow accumulation is actually underesti-mated by AROME–Crocus and SAFRAN–Crocus, and particularly strong accumulations; AROME–Crocus capturing them better yet. These results are in total adequation with the study of Schirmer and Jamieson (2015), who conducted the same work with GEM-LAM (2.5 km resolution NWP model, equivalent to AROME, Erfani et al., 2005) and GEM15 (15 km resolution NWP model, equivalent to
ARPEGE, Mailhot et al., 2006) as atmospheric forcing to SNOWPACK (detailed snowpack model, equivalent to Crocus, Bartelt and Lehning, 2002). They showed the same underestimation of strong accumulations, less marked for the high-resolution forcing. The ETS of GEM-LAM/SNOWPACK for $\Delta SD$ accumulation threshold categories is very close to the ETS shown here for AROME–Crocus.

The comparison with precipitation gauges did not confirm the underestimation of snow accumula-tions since precipitation seemed to be overestimated by AROME, but this paradox can be explained by the uncorrected undercatch of winter precipitation. The assimilation of this data in SAFRAN precipitation analysis tends to reduce them excessively, and subsequently reduce excessively snow accumulations in SAFRAN–Crocus. The problematic assimilation of precipitation gauge measure-
ments in mountainous terrain is also underlined by Schirmer and Jamieson (2015) for the Canadian Precipitation Analysis system CaPA (Mahfouf et al., 2007). This study thus tends to substantiate the idea that variations of SD and SWE measured on the ground could replace precipitation gauges in precipitation analyses in mountainous terrain, as evoked by Schirmer and Jamieson (2015). Mag-nusson et al. (2014) also showed that point SWE data assimilation could improve distributed snow
cover model simulations.

The underestimation of snow accumulation is counterbalanced by an underestimation of the in-tensity of ablation processes. We first showed that wind-induced erosion of the snowpack consti-tuted the major cause of the underestimation of strong ablations: this small-scale process cannot be captured by a kilometric simulation of the snowpack, but it affects the global scores. Secondly,
we showed that the intensity of strong melting is underestimated. This process has several possible sources, that need to be further explored: physical description of melting within the snowpack model, incoming shortwave and longwave radiations in the atmospheric forcing affecting the snowpack sur-face energy balance, formulation of turbulent fluxes. Furthermore, this result presents a paradox with evaluation of Crocus model forced by in-situ meteorological measurements (Brun et al., 1992;





Vionnet et al., 2012), where such a bias has never been noticed. It will be essential to refine the evaluation of the snowpack model in such conditions with the criteria proposed in this paper. Finally, a simultaneous study of SWE and SD evolution gave the opportunity to evaluate the simulated bulk snowpack density. A global underestimation was shown for AROME–Crocus, supporting the hypothesis of an insufficient settling of the snowpack after snowfall in Crocus. This hypothesis is

consistent with previous simulations at Col de Porte station in the Alps (not shown). Consequently, all processes contributing to the decrease of snow depth are underestimated, in a stronger proportion than for accumulations, which leads to a global overestimation of snow depths, through a smoothing of extreme variations. These opposite biases artificially imply a smaller bias for SAFRAN–Crocus than AROME–Crocus. This daily-scale study thus highlights the limitations of global scores (bias,

RMSE, STDE) for a physical quantity like snow depth, integrating several physical processes. Another limitation is the cumulative error during the winter season. Representativeness of stations, affected by local effects, may also be questionned (Grünewald and Lehning, 2015); although the large sample of stations, as well as their large spatial and altitudinal distribution, may reduce the impact of such issues in the present study.

Several limitations also have to be tackled concerning the daily variations of SD and SWE. Data series need to be processed very carefully, since one odd value in the observations would have a double impact in terms of daily variations. Moreover, the daily increase of snow depth not only includes the fresh snowfall but also its own settling and the settling of the underlying layers during one day. This phenomenon would tend to reduce the estimated snow accumulation. A time interval of 6 hours

would be more appropriate following Fischer (2011), but the availability of measurements only made it possible for the automatic stations. $\Delta SWE$ measurements enable to prescind from the issue of snow settling, since it does not affect the snowpack mass. However, SWE measurements by cosmic ray snow gauges are associated with noise due to atmospheric conditions (Gottardi et al., 2013), and thus requires a 24h-median smoothing, which subsequently limits the accuracy of $\Delta SWE$ values

to $\pm 10\%$. Finally, daily variations of snowpack depth or mass are strongly impacted by wind-blown snow events, as shown on Fig. 13: beyond the inherent information about such events, using measurements of snow on the ground for deriving snowfall quantities would require a correction by additional information from snowdrift measurements, as suggested by Fischer (2011).

    These results underline the relevance of AROME–Crocus forecasts to provide high-resolution

spatial patterns of the snowpack in the Pyrenees, while Vionnet et al. (2015b) got similar results in the French Alps. They raise issues about how to combine this promising potential to the assimilation of observations in mountainous terrain, in order to set up a spatially-distributed meteorological analysis system that could substantially improve the atmospheric forcing, as done previously by SAFRAN at the massif scale (Durand et al., 1993). Indeed, most of the uncertainties of a snowpack

simulation come from the atmospheric forcing (Raleigh et al., 2015). To deal with that, the use of complementary observations in complex terrain is necessary, with a particular emphasis on precip-



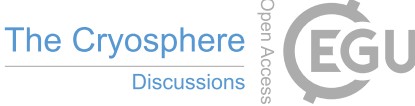

itation. For instance, Birman et al. (2015) recently developed a new precipitation analysis system, combining a priori informations from AROME with ground-based and radar observations. Satellite cloud masks could also be used to improve the incoming radiations (e.g. Hinkelman et al., 2015);

and new polarimetric radar products could help to determine the snow/rain limit (e.g. Augros et al., 2015). The development of higher-resolution versions of AROME or the use of downscaling methods on the meteorological forcing (Vionnet et al., 2015a) would enable sub-kilometric snowpack simulations considering effects of slope and aspect on incoming radiations. It would then allow a direct comparison to satellite images of snow cover. Additionally, observations can also be assimilated

directly within the snowpack model, e.g. as done by Charrois et al. (2015) for optical reflectances in model Crocus. Finally, as all the errors cannot be eliminated, the potential of using ensemble high-resolution forecasts should also be explored in the future. The interest for forecasting extreme hydrological events has been demonstrated (Vié et al., 2011), and Vernay et al. (2015) illustrated the interest of ensemble forecasting for avalanche hazard assessment.

High benefits can also be derived from AROME short-range forecasts: further studies at shorter time scales would shed light on AROME potential for snowpack evolution forecast for high impact events, like intense snowfall triggering off avalanches, rain on snow events or ice layer formation.

*Acknowledgements.* The authors acknowledge Electricité De France and Confederación Hidrográfica del Ebro for providing snow water equivalent and snow depth measurements from their Pyrenean automatic stations

network, Servei Meteorològic de Catalunya for providing snow depth measurements from the automatic weather stations network of Catalunya, S. Gascoin (CESBIO) for providing snow depth measurements from automatic station Bassies, J. Revuelto (Instituto Pirenaico de Ecología) for providing snow depth measurements from automatic station Izas. We particularly thank E. Bazile (CNRM-GAME), F. Gottardi (EDF-DTG), S. Morin (CNRM-GAME/CEN), R. Mott (WSL-SLF) and B. Vincendon (CNRM-GAME) for help and discussions.




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
