# Peer review of "Snowpack modelling in the Pyrenees driven by kilometric resolution meteorological forecasts"

_The Cryosphere, 2016_

## Referee Comment (RC1) · R. L. H. Essery (Referee) · 6 Mar 2016

Queno et al. present an interesting evaluation of high-resolution snowpack simulations in a mountainous region with a reasonably high density of observations. There is quite a lot of overlap in descriptions of the snow model, NWP model and analysis system with a paper by the same authors cited herein (Vionnet et al. 2015b), which addresses some similar issues in a different region. Some repetition will be inevitable to allow the papers to be read independently, but if both are to be published it is the differences between them that will be most interesting. This paper also has a lot of figures, sometimes with quite limited discussion; I think that some consideration could be given to the balance between figures and text.

line 35. Redistribution of snow by avalanches would also be worth mentioning here.

[Figure]

Figure 1. To divide the Pyrenees into western, central and eastern regions, it might seem more obvious to have Haute-Bigorre in the central region and Haute-Ariege and Andorra in the eastern region. Is the division trying to distinguish north-south gradients also? The acronym "SD" is used in the Figure 1 caption but not explained until line 194

216. RMSE is barely mentioned hereafter. Knowing two out of bias, RMSE and STDE, the other one can be determined; what is the point of considering all three?

Rainbow colour schemes, as used in Figures 3 and 4, are deprecated.

Figure 5. The cross section passes close to 2 or 3 precipitation measurement stations on the French side. Could these measurements be shown?

Figure 7. I am not sure that this figure adds anything that is not already clear from Figure 6 and the text in lines 347 to 361.

350. It is not clear what "mechanically counterbalanced" means here.

368-376. There is little interpretation of ETS beyond this paragraph. Are Figure 9 and the associated description of ETS essential to the paper?

403. Contrasting accumulation error to precipitation error is not straightforward because it also involves modelled snow density (discussed later).

420. Crocus has its own index of snow drift. Why is it not used? What difference would it make?

Figure 12 clearly shows that the neglect of wind redistribution of snow in kilometre-scale simulations accounts for some errors in comparison with point-scale measurements, but does this really matter? Apart from snow that sublimates in transit (which Crocus can estimate), the snow removed by wind will just end up in a drift somewhere else, likely within the same model grid cell. Snow redistribution is of course enormously important for loading on avalanche slopes, but that isn't being discussed here.

The English is always good enough to be understood but will require editing to be

perfect. There are several constructions of the type "allows to capture" and "allows to avoid" often used by French authors writing in English; "allows capturing" and "allows avoidance of" or just "captures" and "avoids" would be better English. "deplored" (212) is a rather strong term; "adequation" (532) and "prescind" (581) are English words but very uncommon ones – there will be better choices.

---

## Referee Comment (RC2) · Anonymous Referee #2 · 14 Mar 2016

**1 General comments**

**1.1 Summary of the manuscript**

Queno et al. are evaluating an analysis model and a numerical weather prediction (NWP), respectively, to be able to force a snow cover model at up to 130 stations in the Pyrenees. A time period of four winter seasons was analysed covering very different winters. Their goal was to compare the quality of a 2.5 km resolution NWP model with a coarser analysis system in terms of spatial variability, timing and amount of precipitation, ablation processes and settlement, and amounts of snow depth (HS) and Snow Water Equivalent (SWE). They concluded that the NWP and analysis system produced a positive bias in snow depth, which resulted from an underestimation of accumulation and a larger underestimation of ablation fluxes. Especially large fluxes

were particularly underestimated. For decrease of HS they addressed issues causes by melt, wind and settling separately and concluded that wind erosion was responsible for the largest error during ablation. In general the fine resolution NWP model was found to be better in many analysed aspect.

1.2 Overview of the review

Queno et al. addressed an interesting topic for mountain snow hydrology or avalanche research. Reliable input of solid precipitation and resulting SWE or HS states, is crucial for applications in mountainous terrain, and studies covering long time periods are rare. Also, ablation and densifications processes also interesting to evaluate. The study addressed different error sources, i.e. the meteorological forcing, the snow cover modelling, not included processes, and observation errors. While timing and amount of fluxes and amount of state variables were quantitatively analysed, the conclusion of a better spatial representation by the finer resolution NWP model was analysed qualitatively at one single (but interesting) point in time. The better spatial representation of the NWP forcing is one major conclusion and thus this analysis needs to be enhanced. The impact of the manuscript can be enhanced using different NWP models as meteorological forcing to additional snow cover models with different settlement or melt implementations, which will allow users of those kind of models to choose accordingly. Another increase in impact could be achieved with addressing reasons for errors during melt and settling. Meteorological variables responsible for errors provided by the NWP model or the analysis system could be evaluated, as solar radiation or air temperature. Snow model runs with meteorological weather stations instead of modelled input data would be a solution to discriminate error sources between meteorological forcing or subsequent snow cover modelling.

These suggestions would also decrease the similarity to a cited non-published study including many of the authors of this manuscript (Vionnet et al., 2015b). I think this manuscript is worth publishing despite these similarities after addressing the comments mentioned below.

[Figure]

There are language and spelling issues, so I suggest an accurate editing by a native speaker.

2. Specific comments

2.1 Spatial variability of snow depth

The spatial variability of snow depth was evaluated with Figure 4 in comparison with snow cover fraction at a single point in time. This is indeed an interesting situation, but a more quantitative comparison is needed to conclude that AROME delivers a more realistic spatial variability. First, station observations can be used for this situation of large differences between South and North, pooled in two groups, for example. This would decrease the problem that only snow depth variability and snow cover fraction is compared. Second, one more year can be easily be included. Third, depletion curves can be derived between observed and modeled snow cover fraction.

So far, the authors only discussed precipitation amount differences between SAFRAN and AROME for differences in spatial variability of snow depth. The authors may also comment on differences in precipitation phase or in melt processes, which probably happened repeatedly at lower elevations on the Spanish side.

2.2 Wind erosion major cause for underestimating ablation or decrease in HS

To my opinion it is not clear that wind erosion is the major cause with presented results (line 552). The authors need to be more precise when discussing the data to draw this conclusion. One concern in this regard is that in Figure 13 only a small subset of stations are used, for which wind effects are anticipated. This makes it difficult to compare errors caused by melt or wind erosion.

2.3 Similarity to Vionnet et al. (2015b)

The same model setup was evaluated not in the in the Pyrenees but in the French Alps by Vionnet et al. (2015b). They also evaluated spatial distribution of snowfall similarly to Figure 4 and 5 in this manuscript. They also assessed categorical scores of daily

precipitation. Additional aspects of this manuscript are analysed processes of ablation and settling. This manuscript also uses SWE and HS measurements, additionally to precipitation gauges, to evaluate accumulation. After enhancing the spatial variability part I suggest that this study is publishable additionally to Vionnet et al. (2015b). Other strategies to enhance the impact of this manuscript (see section 1.2) will further discriminate the both studies.

3. Technical comments

Figure 15: Number of observations are missing.

How is the precipitation phase determined? As output from the NWP model and analysis system or with the by the snow model?

The problematic observations of precipitation gauges can be better defined and speculations of the precipitation phase could be reduced if the analysis of Figure 11 would also be performed only for days when snowfall is likely (cold, or dependent on NWP model output).

Why do the authors use the HSS for the evaluation with precipitation gauges and the ETS for snow depth sensors? This reduces the direct comparison between the both evaluation measures.

Line 255: The authors could later provide a summary for reasons causing this high standard deviation error.

Two many figures. I would suggest to delete Figure 12 since there is no additional value shown, and either Figure 16 or 17.

Line 577 and in References: Gruenewald and Lehning (2015) must be 2014.

---

## Author Comment (AC1) · 29 May 2016

**Answer to R. Essery**

We thank R. Essery for his insightful comments. We answered below to all his points. His comments are in bold while our answers appear in normal font. Changes in the manuscript appear in red.

- **Queno et al. present an interesting evaluation of high-resolution snowpack simulations in a mountainous region with a reasonably high density of observations. There is quite a lot of overlap in descriptions of the snow model, NWP model and analysis system with a paper by the same authors cited herein (Vionnet et al. 2015b), which addresses some similar issues in a different region. Some repetition will be inevitable to allow the papers to be read independently, but if both are to be published it is the differences between them that will be most interesting.**

The reviewer is right. There are some similarities between Vionnet et al. (2015b) and our manuscript since they both deal with snowpack modelling issues over mountainous areas using atmospheric forcing from a NWP model. However, the two papers are rather complementary because each one brings a detailed analysis of the spatial variability related to the geographical location of mountains: results over the Alps (discussed in Vionnet et al.) can hardly be generalized to the Pyrenees mountains. Our study focuses on an extended assessment of the quality of snowpack simulations in the Pyrenees, regarding snow depth and SWE point evaluation, snow cover spatial variability, accumulation and ablation processes. On the other hand, Vionnet et al. (2015b) focus firstly on the capabilities of AROME to accurately represent the complex atmospheric variability in the French Alps in wintertime and presents an extended discussion on NWP modelling in complex terrain. Snowpack simulations driven by AROME are then evaluated only against ground-based measurement of snow depth.

Since the snowpack model Crocus and the high resolution NWP model (AROME) are used in both papers, it is quite difficult to avoid redundancies between the two articles which may occur in the description section. We consider that a detailed description of data/models is necessary so that the paper can be read independently. However, we managed to synthesize this section since the atmospheric forecast is not the main focus of this study: the description of AROME physics and data assimilation schemes was deleted, and replaced by a reference to the paper by Seity et al. (2011), which gives a comprehensive description of the AROME model.

--- CHANGES IN MANUSCRIPT (line 152) ---
*A detailed description of the physics and data assimilation schemes can be found in Seity et al. (2011). In particular, the precipitation phase is derived from the cloud microphysical scheme.*

Moreover and in agreement with referee #2 suggestion, a new section dedicated to a quantitative evaluation of simulated snow cover distribution with respect to MODIS snow cover fraction images has been added. It includes a table synthesizing the mean similarity scores by domain and winter, and a figure representing the evolution of daily similarity scores during the winter 2011/2012. A detailed description of these results is provided bellow.
This new study provides new and relevant evidence on the quality of the snow simulations using AROME-Crocus not yet addressed in other publications.

--- CHANGES IN MANUSCRIPT (line 229) ---

[revised manuscript text omitted]

Additionally, the evaluation of precipitation forecast with HSS has been removed, since it did not bring major conclusions to the study.

- **This paper also has a lot of figures, sometimes with quite limited discussion; I think that some consideration could be given to the balance between figures and text.**

We agree with the referee. Figure 7 (cumulated daily SD variations by category) has been removed as suggested in another comment (Figure 6 – 7 after revision – and text were sufficient).

--- CHANGES IN MANUSCRIPT (line 382) ---
*In terms of quantities, the categorical sums of ΔSD (not shown) indicate that SAFRAN-Crocus strongly underestimates the high accumulation quantities.*

Figure 10 (HSS for precipitation) and the associated paragraph have been removed, as explained previously.

Figure 15 (cumulated daily SWE variations by category) has been removed. Overall, daily SWE variations study did not bring new conclusions, compared to the daily SD variations study. Furthermore, despite the 24h-median smoothing, some noise remained in some time series which increased the uncertainty of the values (compared to daily SD variations).

One figure has been added (new section) since it brings relevant information to our study, so the revised version of the paper has two figures less.

- **line 35. Redistribution of snow by avalanches would also be worth mentioning here.**

Redistribution by avalanches has been added, as suggested.

--- CHANGES IN MANUSCRIPT (line 32) ---
*At a smaller scale (less than 100m), processes like wind-induced erosion (Pomeroy and Gray, 1995), avalanches (Schweizer et al., 2003) or preferential deposition of snowfall on the leeward slopes (Lehning et al., 2008), play a decisive role on snow distribution (e.g. Mott et al., 2010).*

- **Figure 1. To divide the Pyrenees into western, central and eastern regions, it might seem more obvious to have Haute-Bigorre in the central region and Haute-Ariege and Andorra in the eastern region. Is the division trying to distinguish north-south gradients also?**

Western, central and eastern regions are defined following the climatological study of Maris et al. (2009), and unpublished studies of CNRM/CEN based on SAFRAN reanalyses. As most of the disturbed flows constituting the snowpack in winter are NW/N flows, this division indeed includes north-south gradients.

- **The acronym "SD" is used in the Figure 1 caption but not explained until line 194**

An explanation of the acronyms SD and SWE has been added in the introduction.

--- CHANGES IN MANUSCRIPT (line 91) ---
*Section 4 details the results following three main axes: (i) global scores and spatial distribution of snow depth (SD); (ii) daily snow depth variations and winter precipitation; and (iii) comparison to snow water equivalent (SWE) scores and study of bulk snowpack density.*

- **216. RMSE is barely mentioned hereafter. Knowing two out of bias, RMSE and STDE, the other one can be determined; what is the point of considering all three?**

The reviewer is right. Only bias and STDE have been kept. RMSE was used shortly in section 4.2.1, it has been replaced by STDE and bias.

--- CHANGES IN MANUSCRIPT (line 207) ---
*Two error metrics were used: the bias and the Standard Deviation Error (STDE, which represents the temporal and spatial dispersion around the bias).*

--- CHANGES IN MANUSCRIPT (line 364) ---
*The STDE of daily $\Delta SD$ indicates the ability of the model to forecast (or analyse) the appropriate daily evolution of snow depth. This score was computed for AROME–Crocus and SAFRAN–Crocus. It is equal to 7 cm (and bias equal to 0 cm) for both models, with low spatial variation. The STDE is slightly higher in the most snowy winters (8 cm in 2012/2013 and 2013/2014 against 6 cm in 2010/2011 and 2011/2012).*

- **Rainbow colour schemes, as used in Figures 3 and 4, are deprecated.**

The colour schemes of these figures have been changed.

- **Figure 5. The cross section passes close to 2 or 3 precipitation measurement stations on the French side. Could these measurements be shown?**

Precipitation measurement stations have not been included for two reasons: (1) they are located at rather low altitudes and cannot discriminate the precipitation phase; (2) the undercatch issue of precipitation gauges. However, we have chosen three SWE measurement stations located close to the transect, and with a similar exposure to the flows as the modelled topography. Cumulated snowfalls have been derived from cumulated positive daily $\Delta SWE$. These measurements (and their actual altitude) have been added to the cross section in Fig. 5, as well as a short comment of these observations in the text.

--- CHANGES IN MANUSCRIPT (line 323) ---
*They are represented in Fig. 5 along a NW/SE cross section, as well as cumulated positive $\Delta SWE$ from measurements of three stations close to the transect.*

--- CHANGES IN MANUSCRIPT (line 333) ---
*AROME simulations are in good agreement with the two Spanish stations, which are located at an altitude close to the model's topography. SAFRAN snowfalls are too low at the station closest to the border, but in good agreement at the second Spanish station. Observations for France are in better agreement with AROME than with SAFRAN, but still higher than both simulations. This may be due to the difference of altitude with the models.*

--- CHANGES IN MANUSCRIPT (caption of Fig. 5) ---
*Cross section of cumulated snowfall from 1 October 2011 to 22 February 2012 for AROME forecasts (blue) and SAFRAN reanalysis (red), with topography plotted on the right axis in grey. Cumulated positive $\Delta SWE$ from measurements of three stations close to the transect are represented with black dots; their actual altitude is represented with black stars. The locations of the transect (red) and stations (blue stars) are given on the upper right map.*

- **Figure 7. I am not sure that this figure adds anything that is not already clear from Figure 6 and the text in lines 347 to 361.**

Figure 7 presented the differences in terms of quantities, but the text associated to Figure 6 (Fig. 7 after revision) may be sufficient. Figure 7 was removed, as suggested.

--- CHANGES IN MANUSCRIPT (line 382) ---
*In terms of quantities, the categorical sums of ΔSD (not shown) indicate that SAFRAN– Crocus strongly underestimates the high accumulation quantities.*

- **350. It is not clear what "mechanically counterbalanced" means here.**

The word "mechanically" was ambiguous and has been removed. A brief explanation has been added.

--- CHANGES IN MANUSCRIPT (line 385) ---
*It is counterbalanced by an overestimation of small accumulation quantities, since an underestimated strong accumulation event is counted in the smaller accumulation category.*

- **368-376. There is little interpretation of ETS beyond this paragraph. Are Figure 9 and the associated description of ETS essential to the paper?**

We used the ETS in the paper since it provides an overview of models skills. Figure 9 shows that SAFRAN-Crocus and AROME-Crocus have equivalent scores in terms of daily snow depth accumulation, AROME-Crocus being better for accumulations higher than 10 cm/day. The ETS has also been used by Schirmer and Jamieson (2015) for evaluating snow accumulations simulated by equivalent models (GEM-LAM/SNOWPACK). A direct comparison of both models configurations is thus possible. For these two reasons, we decided to keep the ETS in the revised version of the paper.

- **403. Contrasting accumulation error to precipitation error is not straightforward because it also involves modelled snow density (discussed later).**

This issue has been mentioned in the revised version of the manuscript.

--- CHANGES IN MANUSCRIPT (line 437) ---
*The difference between accumulation and precipitation errors also involves modelled snow density: this issue is discussed in section 4.3.*

- **420. Crocus has its own index of snow drift. Why is it not used? What difference would it make?**

R. Essery points out an interesting way to detect wind erosion. Crocus has indeed a snow drift index derived from surface wind (given by the input forcing) and a mobility index based on the modelled snow surface properties (Guyomarc'h and Mérindol, 1998). Such an index makes it possible to take into account snowpack properties additionally to wind speed in the determination of blowing snow occurrence. However, we have chosen not to use this index for several reasons.

Firstly, if we consider the index coming from our snowpack simulations, it is computed using modelled snow surface properties and 10-m wind as simulated by the atmospheric forcing (e.g. AROME). Simulated wind at 2.5-km grid spacing in mountainous terrain can largely differ from observed wind due to topographic features (ridges, depressions…) not reproduced at 2.5-km grid spacing. Using the AROME wind at the Maupas station would for example

give wind erosion detection almost every day in winter, because the simulated wind is too strong.

Then, a solution would be to use wind observed at the automatic stations combined with simulated snow surface properties to derive a new snowdrift index. We computed this index (cumulated for each day) at Maupas station for winter 2012/2013. It is represented on Fig. R1-1 as black bars, together with the BSD detection (green). Both detections are in fair agreement, particularly for the strongest events.

However, we consider that taking into account the modelled surface properties does not necessarily improve the detection of blowing snow events. Indeed, snow surface properties have been modified by previous non-simulated wind erosion events, since there is no ablation by wind transport in Crocus. For instance, a 60-cm decrease of snow depth occurs in mid-December: the snowpack is totally different after this event which is not simulated; the subsequent snowdrift index may be far from reality.

Consequently, in the revised version of the manuscript, the wind erosion index is only based on observed wind and simulated melting.

[Figure]

Figure R1-1: Snow depth simulated by AROME-Crocus (blue) and observed (black) at Maupas station, 2012/2013. BSD are identified in green and cumulated daily snowdrift index is represented by black bars.

- **Figure 12 clearly shows that the neglect of wind redistribution of snow in kilometrescale simulations accounts for some errors in comparison with point-scale measurements, but does this really matter? Apart from snow that sublimates in transit (which Crocus can estimate), the snow removed by wind will just end up in a**

**drift somewhere else, likely within the same model grid cell. Snow redistribution is of course enormously important for loading on avalanche slopes, but that isn't being discussed here.**

We thank R. Essery for this comment. Figure 12 (Fig. 11 after revision) is presented in this paper to illustrate how the computation of SD and SWE scores is affected by the occurrence of wind-induced snow transport at stations measuring SD and SWE. This figure clearly shows that wind redistribution strongly impacts observations used for validating the simulations and this must be kept in mind when discussing model results. We then totally agree with R. Essery concerning the fact that wind-induced snow redistribution cannot be represented on a regular grid at 2.5-km grid spacing. Snow redistribution by wind indeed occurs very likely within each grid cell. In the discussion part of the revised version of the paper, we now mention more clearly these two different points.

--- CHANGES IN MANUSCRIPT (line 584) ---
*We first showed that wind-induced erosion of the snowpack constituted the major cause of the underestimation of strong ablations at seven high altitude stations. This small-scale process cannot be captured by a kilometric simulation of the snowpack, since snow redistribution by wind occurs very likely within each grid cell. But the computation of SD and SWE scores is affected by the occurrence of wind-induced snow transport at stations. The impact of blowing snow could not be estimated at all stations. It is probably less significant at lower altitudes.*

- **The English is always good enough to be understood but will require editing to be perfect. There are several constructions of the type "allows to capture" and "allows to avoid" often used by French authors writing in English; "allows capturing" and "allows avoidance of" or just "captures" and "avoids" would be better English. "deplored" (212) is a rather strong term; "adequation" (532) and "prescind" (581) are English words but very uncommon ones – there will be better choices.**

The new version of the manuscript has been edited by a native speaker.

---

## Author Comment (AC2) · 29 May 2016

**Answer to Referee #2**

We thank the referee for his insightful comments. We answered below to all his points. His comments are in bold while our answers appear in normal font. Changes in the manuscript appear in red.

**1 General comments**

**1.1 Summary of the manuscript**

**Queno et al. are evaluating an analysis model and a numerical weather prediction (NWP), respectively, to be able to force a snow cover model at up to 130 stations in the Pyrenees. A time period of four winter seasons was analysed covering very different winters. Their goal was to compare the quality of a 2.5 km resolution NWP model with a coarser analysis system in terms of spatial variability, timing and amount of precipitation, ablation processes and settlement, and amounts of snow depth (HS) and Snow Water Equivalent (SWE). They concluded that the NWP and analysis system produced a positive bias in snow depth, which resulted from an underestimation of accumulation and a larger underestimation of ablation fluxes. Especially large fluxes were particularly underestimated. For decrease of HS they addressed issues causes by melt, wind and settling separately and concluded that wind erosion was responsible for the largest error during ablation. In general the fine resolution NWP model was found to be better in many analysed aspect.**

**1.2 Overview of the review**

**Queno et al. addressed an interesting topic for mountain snow hydrology or avalanche research. Reliable input of solid precipitation and resulting SWE or HS states, is crucial for applications in mountainous terrain, and studies covering long time periods are rare. Also, ablation and densifications processes also interesting to evaluate. The study addressed different error sources, i.e. the meteorological forcing, the snow cover modelling, not included processes, and observation errors. While timing and amount of fluxes and amount of state variables were quantitatively analysed, the conclusion of a better spatial representation by the finer resolution NWP model was analysed qualitatively at one single (but interesting) point in time. The better spatial representation of the NWP forcing is one major conclusion and thus this analysis needs to be enhanced.**

We thank the referee for his suggestions. We have chosen to increase the impact of the manuscript with a more extensive study of the snow cover spatial distribution. This study is detailed in the answer to the specific comment 2.1.

**The impact of the manuscript can be enhanced using different NWP models as meteorological forcing to additional snow cover models with different settlement or melt implementations, which will allow users of those kind of models to choose accordingly. Another increase in impact could be achieved with addressing reasons for errors during melt and settling.**

This study focuses on the use of kilometric resolution NWP models as meteorological forcing to a snowpack model. Over our domain of study (the Pyrenees), only the AROME model is available.

Concerning the snowpack model, we work with Crocus, because our study is performed with a view to operational avalanche forecasting. The aim of this study is not to discuss the quality of results depending on the complexity of the snowpack model. Furthermore, addressing the reasons of the errors during melting and settling requires an extensive study of Crocus physics formulations, which would go beyond the scope of this paper.

**Meteorological variables responsible for errors provided by the NWP model or the analysis system could be evaluated, as solar radiation or air temperature.**

An extensive evaluation of meteorological variables forecasted by AROME in alpine terrain has been already performed by Vionnet et al. (2015b). We have decided to keep the focus of the manuscript on snowpack modelling, through the assessment of snowpack-related variables only. In the manuscript, we also refer to the uncertainties due to the meteorological forcing (precipitations for snow accumulation, incoming radiations for melting...).

**Snow model runs with meteorological weather stations instead of modelled input data would be a solution to discriminate error sources between meteorological forcing or subsequent snow cover modelling.**

We thank the referee for this comment. Using meteorological weather stations as input to the snowpack model is indeed an interesting way to discriminate error sources. However, there is no station in the Pyrenees providing all the measurements necessary for the atmospheric forcing of Crocus. The only station suitable for such a study in the French mountains is the Col de Porte located in the French Alps.

**These suggestions would also decrease the similarity to a cited non-published study including many of the authors of this manuscript (Vionnet et al., 2015b). I think this manuscript is worth publishing despite these similarities after addressing the comments mentioned below.**

**There are language and spelling issues, so I suggest an accurate editing by a native speaker.**

The new version of the manuscript has been edited by a native speaker.

**2. Specific comments**

**2.1 Spatial variability of snow depth**

**The spatial variability of snow depth was evaluated with Figure 4 in comparison with snow cover fraction at a single point in time. This is indeed an interesting situation, but a more quantitative comparison is needed to conclude that AROME delivers a more realistic spatial variability. First, station observations can be used for this situation of large differences between South and North, pooled in two groups, for example. This would decrease the problem that only snow depth variability and snow cover fraction is**

**compared. Second, one more year can be easily be included. Third, depletion curves can be derived between observed and modeled snow cover fraction.**

**So far, the authors only discussed precipitation amount differences between SAFRAN and AROME for differences in spatial variability of snow depth. The authors may also comment on differences in precipitation phase or in melt processes, which probably happened repeatedly at lower elevations on the Spanish side.**

We thank the referee for this very relevant comment. The section dealing with the spatial variability of snow cover has been updated as suggested by the referee. We have completed the study which only described initially a single date of winter 2011/2012. In the revised version of the paper, we present a more quantitative study of AROME-Crocus and SAFRAN-Crocus representation of the snow distribution, through comparisons to MODIS snow cover images during two winters (2011/2012 and 2012/2013). Two new scores have been used to evaluate how simulated snow cover agrees with the MODIS satellite images: the Average Symmetric Surface Distance (average distance from one snow line to the other) and the Jaccard index (evaluating surfaces matching). Both are presented in the manuscript because the ASSD describes more the correspondence of snow lines while the Jaccard index is more representative of the total areas. We get the same results with the two metrics: AROME-Crocus better represents the snow cover distribution than SAFRAN-Crocus. The new section includes a table synthesizing the mean similarity scores by domain and winter, and two figures representing the evolution of daily similarity scores during winter 2011/2012.

--- CHANGES IN MANUSCRIPT (line 229) ---

[revised manuscript text omitted]

**2.2 Wind erosion major cause for underestimating ablation or decrease in HS**

**To my opinion it is not clear that wind erosion is the major cause with presented results (line 552). The authors need to be more precise when discussing the data to draw this conclusion. One concern in this regard is that in Figure 13 only a small subset of stations are used, for which wind effects are anticipated. This makes it difficult to compare errors caused by melt or wind erosion.**

In order to better highlight the contribution of wind erosion to strong decreases of snow depth observed at these seven stations, we have added a quantitative discussion of the results exposed in Fig. 13 (Fig. 12 after revision). We show that wind erosion constitutes 71% of high decreasing rates. There is no overlap of blowing snow days (BSD) with melting snow days (MSD), which means melting is part of the 29% remaining.

For the sake of clarity, we have plotted BSD (instead of all days excluding BSD) on Fig. 13 (Fig. 12 after revision). The same representation has been chosen for MSD in Fig. 14 (Fig. 13 after revision). Similarly to the BSD study, we have shown that MSD represented 42% of high decreasing rates at all stations.

Concerning the smaller subset of stations used for wind erosion study, this is due to the fact that only automatic stations measure wind speed. We have shown that wind erosion is the major cause for underestimating strong ablations for these seven stations located at high altitude (mean altitude: 2203 m.a.s.l). The referee is right that we don't have enough data to conclude that it is the major cause at all stations. Indeed, the contribution of blowing snow may be less significant at lower altitudes. We have qualified this assertion in the discussion.

--- CHANGES IN MANUSCRIPT (line 457) ---
*To quantify the impact of wind-blown snow events on the performance of models, the cumulated ΔSD for AROME–Crocus and observations are plotted in Fig. 12, for BSD and all days, with a finer categorization of SD decreases. This study is restricted to seven automatic stations measuring wind speed and SD (mean altitude: 2203 m.a.s.l). For observations, BSD contribute to all decreasing rates, in the strongest proportion for high decreasing rates (less than -20 cm). For AROME–Crocus, BSD do not contribute to the strong ablation categories but to small ablation and accumulation categories in the same proportions. Cumulated ΔSD for high decreasing rates is equal to -1106 cm for all observations, and equal to -781 cm for BSD only (excluding MSD), while it is equal to 0 cm for AROME–Crocus in both cases. It means that wind-blown snow is the main contributor (71%) to this category, the remaining contribution coming from MSD or other processes.*
*Similarly, the cumulated ΔSD is plotted in Fig. 13 for MSD and all days, at all SD stations. Very strong melting (more than 20 cm.day$^{-1}$) is seldom observed, but never predicted. Strong melting (between -20 20 cm.day$^{-1}$ and -10 20 cm.day$^{-1}$) is much under-represented by models, while melting of less than 10 20 cm.day$^{-1}$ is over-represented. Cumulated ΔSD for high decreasing rates (more than 20 cm.day$^{-1}$) is equal to -7741 cm for all observations, and equal to -3215 cm for MSD only, while it is equal to -41 cm for AROME–Crocus in both cases. Melting snow represents 42% of this category, the remaining contribution coming from BSD or other processes. The behaviour of SAFRAN–Crocus is similar to AROME–Crocus for BSD and MSD (not shown). The simple diagnostics of BSD and MSD may miss some blowing-snow or melting events.*
*Consequently, the underestimation of strong decreasing rates comes mainly from ablation processes: on the one hand, from wind-blown snow events which are not represented by models, as they are small scale processes; and on the other hand, from an underestimation of strong snowpack melting (more than 10 cm.day$^{-1}$). Other reasons for very high decreasing rates can be the strong settling after an intense snowfall or a rain-on-snow event, but it probably constitutes a limited part of this category.*

--- CHANGES IN MANUSCRIPT (line 584) ---

*We first showed that wind-induced erosion of the snowpack constituted the major cause of the underestimation of strong ablations at seven high altitude stations. This small-scale process cannot be captured by a kilometric simulation of the snowpack, since snow redistribution by wind occurs very likely within each grid cell. But the computation of SD and SWE scores is affected by the occurrence of wind-induced snow transport at stations. The impact of blowing snow could not be estimated at all stations. It is probably less significant at lower altitudes.*

**2.3 Similarity to Vionnet et al. (2015b)**

**The same model setup was evaluated not in the in the Pyrenees but in the French Alps by Vionnet et al. (2015b). They also evaluated spatial distribution of snowfall similarly to Figure 4 and 5 in this manuscript. They also assessed categorical scores of daily precipitation. Additional aspects of this manuscript are analysed processes of ablation and settling. This manuscript also uses SWE and HS measurements, additionally to precipitation gauges, to evaluate accumulation. After enhancing the spatial variability part I suggest that this study is publishable additionally to Vionnet et al. (2015b). Other strategies to enhance the impact of this manuscript (see section 1.2) will further discriminate the both studies.**

The reviewer is right. R. Essery in his review pointed out the same aspect and we reproduce below the answer that we gave to R. Essery.

There are some similarities between Vionnet et al. (2015b) and our manuscript since they both deal with snowpack modelling issues over mountainous areas using atmospheric forcing from a NWP model. However, the two papers are rather complementary because each one brings a detailed analysis of the spatial variability related to the geographical location of mountains: results over the Alps (discussed in Vionnet et al.) can hardly be generalized to the Pyrenees mountains. Our study focuses on an extended assessment of the quality of snowpack simulations in the Pyrenees, regarding snow depth and SWE point evaluation, snow cover spatial variability, accumulation and ablation processes. On the other hand, Vionnet et al. (2015b) focus firstly on the capabilities of AROME to accurately represent the complex atmospheric variability in the French Alps in wintertime and presents an extended discussion on NWP modelling in complex terrain. Snowpack simulations driven by AROME are then evaluated only against ground-based measurement of snow depth.

Since the snowpack model Crocus and the high resolution NWP model (AROME) are used in both papers, it is quite difficult to avoid redundancies between the two articles which may occur in the description section. We consider that a detailed description of data/models is necessary so that the paper can be read independently. However, we managed to synthesize this section since the atmospheric forecast is not the main focus of this study: the description of AROME physics and data assimilation schemes was deleted, and replaced by a reference to the paper by Seity et al. (2011), which gives a comprehensive description of the AROME model.

--- CHANGES IN MANUSCRIPT (line 152) ---
*A detailed description of the physics and data assimilation schemes can be found in Seity et al. (2011). In particular, the precipitation phase is derived from the cloud microphysical scheme.*

Like the focus on accumulation and ablation processes, the study of snow cover spatial distribution of simulations vs MODIS images (Fig. 4) is specific to our paper. Cross sections of simulated snowfalls (Fig. 5) are also presented by Vionnet et al. (2015b), but it is used in the present paper as an interpretation of the differences of snow cover distribution between AROME-Crocus, SAFRAN-Crocus and MODIS.

Additionally, the evaluation of precipitation forecast with HSS has been removed, since it did not bring major conclusions to the study.

Following the referee's suggestion, a new quantitative study of spatial variability has been added (described previously).

**3. Technical comments**

**Figure 15: Number of observations are missing.**

Following a comment of R. Essery, Figure 15 has been removed to compensate the increasing number of figures due to the new section about spatial variability and to improve the balance between figures and text in this paper. Overall, daily SWE variations study did not bring new conclusions, compared to the daily SD variations study. Furthermore, despite the 24h-median smoothing, some noise remained in some time series which increased the uncertainty of the values (compared to daily SD variations).

**How is the precipitation phase determined? As output from the NWP model and analysis system or with the by the snow model?**

Snowfall and rainfall are distinguished as outputs of the NWP model (from the cloud microphysical scheme) and the analysis system (threshold $T_{2m}=1°C$). A mention of this issue has been added in the description of models.

--- CHANGES IN MANUSCRIPT (line 153) ---
*In particular, the precipitation phase is derived from the cloud microphysical scheme.*

--- CHANGES IN MANUSCRIPT (line 177) ---
*The precipitation phase is derived from a simple threshold of 1°C air temperature at 2 m above the ground.*

**The problematic observations of precipitation gauges can be better defined and speculations of the precipitation phase could be reduced if the analysis of Figure 11 would also be performed only for days when snowfall is likely (cold, or dependent on NWP model output).**

A brief mention of the effect of wind on snowflakes trajectories has been added, in supplement to the reference to literature, which seems sufficient for further details.

--- CHANGES IN MANUSCRIPT (line 432) ---
*The undercatch of solid precipitations by gauges, mainly due to wind effects on falling snowflakes trajectories, is well known and very variable. This issue is investigated by the WMO Solid Precipitation InterComparison Experiment (e.g. Wolff et al., 2015).*

As suggested by the referee, the analysis of Figure 11 (Fig. 10 after revision) has been restricted to "cold" days (i.e. daily maximal 2m-temperature lower than 2°C), when snowfall is more likely (rainfall now represents only 6% of total AROME precipitation). With this criterion, the study period has been extended from October to June (instead of December-April).

--- CHANGES IN MANUSCRIPT (line 413) ---
*A complementary information on winter precipitation comes from the network of gauges in the French Pyrenees (red dots in Fig. 1). Daily accumulations of precipitation (rainfall plus snowfall, cumulated from 6UTC to 6UTC) from the forcing models are then directly compared to precipitation gauges measurements, for days with a maximum temperature of 2°C in order to reduce the proportion of rainfall amongst precipitation. Most of these observations are assimilated in SAFRAN reanalyses, while they are not taken into account in AROME forecasts. Figure 10 shows cumulated precipitation by category for both models and observations (right) compared to cumulated ΔSD at the same stations (left). Contrary to ΔSD, AROME overestimates precipitation measured by gauges (+ 73 %). The optimal interpolation basis of the SAFRAN analysis system should mathematically not be biased on the assimilated observations over a long period. The slightly positive bias obtained in this study (+ 17 %) may be linked to the fact that some assimilated observations are not included in our evaluation dataset and/or to differences between the climatological guess and the mean precipitation amount of the 4 years under study. The strong overestimation of AROME is particularly notable for the largest amounts. The different distribution of precipitation and ΔSD for AROME, with a higher proportion of strong precipitation than of strong snow accumulations, may be due to settling effects: the stronger the snowfall, the stronger the snowpack settles under its own mass, which shifts the distribution to the left.*

**Why do the authors use the HSS for the evaluation with precipitation gauges and the ETS for snow depth sensors? This reduces the direct comparison between the both evaluation measures.**

We agree with the referee that using two different scores could bring some confusion. The HSS was used for precipitation evaluation in order to facilitate comparisons with other NWP precipitation evaluations, and particularly with AROME precipitation evaluation in the French Alps by Vionnet et al. (2015b). The ETS was used for daily snow depth variations evaluation in order to allow direct comparison with the categorical study of snow accumulations by Schirmer and Jamieson (2015). This comparison is particularly relevant since equivalent NWP and snowpack models were used (GEM-LAM and SNOWPACK).

Following a comment of R. Essery concerning the balance between figures and text in the paper, the evaluation of precipitation through the HSS has been removed since it did not bring major conclusions to the article.

**Line 255: The authors could later provide a summary for reasons causing this high standard deviation error.**

The STDE represents the temporal (within a season and between seasons) and spatial (between stations) dispersion around the bias. The underestimation of the intensity of daily snow depth variations may explain a high STDE: daily variations are not well reproduced which implies a daily variation of the bias, and thus a higher dispersion. This issue has been mentioned in the discussion as suggested by the referee.

--- CHANGES IN MANUSCRIPT (line 601) ---
*Consequently, all processes contributing to the decrease of the snow depth are underestimated, in a stronger proportion than for accumulations, which leads to a global overestimation of snow depths, through a smoothing of extreme variations. These opposite biases artificially imply a smaller bias for SAFRAN--Crocus than for AROME--Crocus. The underestimation of the intensity of daily variations also implies daily variations of the bias, hence a high dispersion around the mean bias, which partly explains a high STDE.*

**Two many figures. I would suggest to delete Figure 12 since there is no additional value shown, and either Figure 16 or 17.**

We agree with the reviewer. Figure 7 (cumulated daily SD variations by category) has been removed as suggested by R. Essery (Figure 6 – 7 after revision – and text were sufficient).

--- CHANGES IN MANUSCRIPT (line 382) ---
*In terms of quantities, the categorical sums of ΔSD (not shown) indicate that SAFRAN-Crocus strongly underestimates the high accumulation quantities.*

Figure 10 (HSS for precipitation) and the associated paragraph have been removed, as explained previously.

Figure 15 has been removed as explained previously.

One figure has been added (new section), so the revised version of the paper has two figures less.

**Line 577 and in References: Gruenewald and Lehning (2015) must be 2014.**

The article was first published online in 2014, but the actual date of publication is 2015: http://onlinelibrary.wiley.com/doi/10.1002/hyp.10295/abstract

---

## Author Response (AR2)

**Answer to the Editor and R. Essery**

We thank the Editor for this report, and R. Essery for his new insightful comments. We addressed them below. His comments are in bold while our answers appear in normal font. First revision changes in the manuscript appear in red and second revision changes appear in blue.

**My comments have been thoroughly addressed. In the added material, I think it would be more clear to simply say that J is the number of pixels that are snow covered in both A and B divided by the total number of pixels that are in both A and B.**

A clearer description of the Jaccard index has been added to the formula in the new version.

--- CHANGES IN MANUSCRIPT (line 232) ---
*If A and B represent the simulated and the observed snow cover domain, respectively, J is the number of pixels that are snow covered in both A and B divided by the total number of pixels in the union of A and B:*

$$J = \frac{|A \cap B|}{|A \cup B|}$$

*J is thus dependent on the whole snow covered area. It ranges from 0 to 1, where 0 means no overlap of A and B surfaces, and 1 means A = B.*

**I can follow what MDHD means but think that it would be hard to calculate; is there a package available?**

The MDHD and ASSD, as well as the Jaccard index, are calculated with the Python medpy.metric.binary program from MedPy package, available on: https://pypi.python.org/pypi/MedPy/
This package has been mentioned in the new version.

--- CHANGES IN MANUSCRIPT (line 229) ---

[revised manuscript text omitted]